# TGB-Seq Benchmark: Challenging Temporal GNNs with Complex Sequential Dynamics

**Lu Yi[1], Jie Peng[1], Yanping Zheng[1]\*, Fengran Mo[2], Zhewei Wei[1]\***
[1]Renmin University of China, [2]Université de Montréal
{yilu, peng_jie, zhengyanping, zhewei}@ruc.edu.cn
fengran.mo@umontreal.ca

**Yuhang Ye[3], Zixuan Yue[3]**
[3]Huawei Poisson Lab, Huawei Technology Ltd.
{yeyuhang,yuezixuan}@huawei.com

**Zengfeng Huang[4]**
[4]Fudan University, Shanghai Innovation Institute
huangzf@fudan.edu.cn

## Abstract

Future link prediction is a fundamental challenge in various real-world dynamic systems. To address this, numerous temporal graph neural networks (temporal GNNs) and benchmark datasets have been developed. However, these datasets often feature excessive repeated edges and lack complex sequential dynamics, a key characteristic inherent in many real-world applications such as recommender systems and "Who-To-Follow" on social networks. This oversight has led existing methods to inadvertently downplay the importance of learning sequential dynamics, focusing primarily on predicting repeated edges.

In this study, we demonstrate that existing methods, such as GraphMixer and DyG-Former, are inherently incapable of learning simple sequential dynamics, such as "a user who has followed OpenAI and Anthropic is more likely to follow AI at Meta next." Motivated by this issue, we introduce the Temporal Graph Benchmark with Sequential Dynamics (TGB-Seq), a new benchmark carefully curated to minimize repeated edges, challenging models to learn sequential dynamics and generalize to unseen edges. TGB-Seq comprises large real-world datasets spanning diverse domains, including e-commerce interactions, movie ratings, business reviews, social networks, citation networks and web link networks. Benchmarking experiments reveal that current methods usually suffer significant performance degradation and incur substantial training costs on TGB-Seq, posing new challenges and opportunities for future research. TGB-Seq datasets, leaderboards, and example codes are available at https://tgb-seq.github.io/.

## 1 Introduction

Future link prediction (Divakaran & Mohan, 2020) is a fundamental challenge in various real-world dynamic systems, such as social networks (Daud et al., 2020), e-commerce (Bai et al., 2020), financial systems (Rajput & Singh, 2022). For instance, an online shopping website must decide which items to recommend to users based on their click history, while a social networking platform needs to identify which users may be interested in connecting based on their existing relationships. Among the various approaches for future link prediction, temporal Graph Neural Networks (GNNs) are particularly notable for their flexibility in modeling diverse applications and their representation learning capabilities (Zheng et al., 2024; Skarding et al., 2021; Kazemi et al., 2020). Recently, several temporal GNN methods (Yu et al., 2023) have demonstrated impressive performance in future link prediction on existing benchmarks (Poursafaei et al., 2022). However, most existing datasets are not derived from real-world recommender systems, despite recommendations being a natural and essential application of future link prediction.

---

\*Yanping Zheng and Zhewei Wei are the corresponding authors.

**Observations.** To assess the capability of current temporal GNNs in recommendation tasks, we evaluate their performance on future link prediction using two widely used recommendation datasets, including the user-product interaction network Taobao (Zhu et al., 2018) and the business review network Yelp [1]. Figure 1 presents the performance of three state-of-the-art temporal GNN approaches across these datasets, including EdgeBank (Poursafaei et al., 2022), GraphMixer (Cong et al., 2023) and DyGFormer (Yu et al., 2023). We split these datasets chronologically and randomly sample 100 negative destination nodes for each positive instance, utilizing the Mean Reciprocal Rank (MRR) as the evaluation metric. Besides, we also include SGNN-HN (Pan et al., 2020),

one of the state-of-the-art methods for sequential recommendation to compare with temporal GNNs. Intuitively, these recommendation datasets are comparable to existing datasets (e.g., Wikipedia and Reddit), as all represent typical dynamic systems, and thus, temporal GNNs are expected to perform in a similar trend on these recommendation datasets. However, Figure 1 shows that temporal GNN methods present significant performance degradation compared to their strong results on two previously established datasets and present a substantial performance gap compared with SGNN-HN, which contradicts our intuition. This arises a critical question:

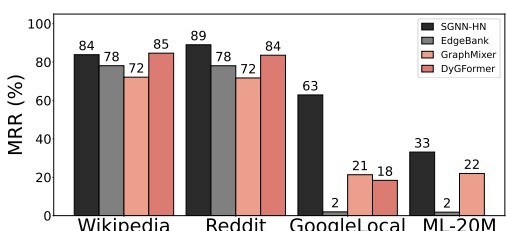

Figure 1: The MRR scores of three selected temporal GNNs and SGNN-HN on two existing datasets (Wikipedia, Reddit) and two recommendation datasets (Yelp and Taobao).

*Why do existing temporal GNN methods, which demonstrate superior performance on existing temporal graph datasets, fail to perform well in a typical downstream application, i.e., recommendation?*

We conjecture that this is because existing datasets, e.g., Wikipedia and Reddit, contain excessive repetitions of historical edges compared to these evaluated recommendation datasets. Consequently, temporal GNNs tend to predict these repeated historical edges via memorizing or aggregating historical edges and perform well on existing datasets. To validate our assumption, we use existing temporal GNNs to predict both repeated and unseen edges and report their MRR scores separately across four widely used datasets: Wikipedia, Reddit, Social Evo. and Enron, following the experimental settings of Figure 1. The results in Figure 2 indicate a substantial prediction performance gap between historical and unseen edges, with differences reaching up to eightfold. This phenomenon implies that existing methods are effective on graphs dominated by repeated edges but fail to generalize to those that emphasize unseen edges. The underlying reason is probably that existing methods tend to rely on the information of historical neighbors, which limits their generalizability. Thus, they can only associate query nodes with their historical neighbors but fail on unseen edges.

**Motivations.** However, future links are typically not simple repetitions of historical ones in many real-world dynamic systems. Instead, the evolution of many dynamic systems often exhibits intricate *sequential dynamics*. For example, on an e-commerce platform, an *entity* (i.e., a customer) who has purchased a smartphone and a phone case is likely to buy a screen protector next. In this context, future interactions of entities typically involve new purchases rather than simply repeating past ones. Therefore, a model must capture the inherent sequential dynamics in these systems to accurately predict future links. *Capturing sequential dynamics involves modeling the evolution of the intentions of entities based on their historical interactions and forecasting unseen interactions.* However, we find that existing temporal GNNs struggle to effectively capture even simple sequential dynamics that exclude repeated edges, despite these dynamic patterns being frequently present in the training set. The observed cases are provided in later Section 3 and Figure 3. On the other hand, existing datasets often contain an excessive number of repeated edges, which undermines the critical aspect of complex sequential dynamics. Evaluating temporal GNN models solely based on these datasets cannot adequately assess their ability to capture complex sequential dynamics.

**Contributions.** To address this gap, we present the Temporal Graph Benchmark with Sequential Dynamics (TGB-Seq), a collection of new benchmark datasets designed to evaluate the systems' ability to capture complex sequential dynamics. TGB-Seq includes four widely-used recommendation datasets and four non-bipartite datasets derived from typical future link prediction scenarios that inherently exhibit complex sequential dynamics, including a movie rating network (ML-20M), an

---

[1]https://www.yelp.com/dataset

Figure 2: The MRR scores of eight popular temporal GNNs for predicting repeated historical edges on four previously established datasets. "Unseen" denotes the performance of unseen edges.

e-commerce interaction network (Taobao), two business review networks (Yelp and GoogleLocal), two "Who-To-Follow" social networks (Flickr and YouTube), a citation network (Patent) and a web link network (WebLink). TGB-Seq datasets are carefully curated to minimize repeated edges. Only Yelp and Taobao contain a small number of repeated edges with the natural behavior that users potentially review or click items multiple times. All TGB-Seq datasets are ensured with medium to large scale toward the practical situation, comprising millions to tens of millions of edges. Overall, we make the following contributions in this paper:

- We demonstrate that existing temporal GNNs fail to capture sequential dynamics in temporal graphs, limiting their generalizations to various real-world scenarios.

- We propose TGB-Seq, a collection of eight benchmark datasets for future link prediction, carefully curated from diverse application domains with intricate sequential dynamics. TGB-Seq focuses on evaluating temporal GNNs' capability to capture sequential dynamics and generalize to unseen edges, addressing the limitations of existing datasets that contain excessive repetitions of edges and overlook the intricate sequential dynamics present in real-world dynamic systems.

- Comprehensive evaluations on TGB-Seq reveal that existing temporal GNNs experience substantial performance declines compared to their impressive results on existing benchmarks. This observation underscores the limited ability of existing methods to capture complex sequential dynamics and demonstrate the distinguishing functionality of TGB-Seq in evaluating such ability.

- We provide a Python package available via pip, enabling seamless dataset downloading, negative sample generation, and evaluation. All code is publicly available on the TGB-Seq GitHub repository. Researchers can submit their methods and compare performance on the TGB-Seq website, where we maintain the leaderboards for all datasets.

## 2 RELATED WORK

**Temporal Graph Datasets and Benchmarks.** Several studies (Poursafaei et al., 2022; Huang et al., 2024b) have highlighted that existing benchmarks for dynamic graph learning often lead to overly optimistic assessments of current approaches. Specifically, widely used datasets such as Reddit, Wikipedia, MOOC, and LastFM (Kumar et al., 2019) suffer from inconsistent preprocessing and simplistic negative sampling strategies, which result in inflated performance metrics and unreliable comparisons. To address these issues, BenchTeMP (Huang et al., 2024a) provides a unified evaluation framework with consistent datasets and comprehensive performance metrics. Poursafaei et al. (2022) construct six dynamic graph datasets across diverse domains, such as politics, economics, and transportation, and introduce two negative sampling strategies to create more challenging evaluations. TGB (Huang et al., 2024b) further advances the field by introducing several large datasets for future link prediction, establishing a comprehensive benchmark based on the unified evaluation metric of MRR. Gastinger et al. (2024) extend TGB with multi-relational datasets for Temporal Knowledge Graphs (TKG) and Temporal Heterogeneous Graphs (THG). These extensions focuses on future link prediction on large-scale TKG and THG datasets.

**Temporal Graph GNNs for future link prediction.** Future link prediction is a critical task in various dynamic systems, which aim to predict future interactions or relationships between entities based on historical data. To capture the evolution pattern, memory-based methods such as TGN (Rossi et al., 2020), Jodie (Kumar et al., 2019), DyRep (Trivedi et al., 2019), and APAN (Wang et al., 2021c), use dynamic memory modules to store and update node information during interactions, allowing for

more effective modeling. On the other hand, approaches like TGAT (Xu et al., 2020), CAWN (Wang et al., 2021d), TCL (Wang et al., 2021a), GraphMixer (Cong et al., 2023), and DyGFormer (Yu et al., 2023), aggregate historical neighbor information directly during prediction without memory modules. These methods employ contrastive learning and Transformer-based techniques to capture evolving node interactions and temporal dependencies. The Hawkes process (Hawkes, 1971; Mei & Eisner, 2017) is another widely used technique for capturing the impacts of historical events on current events and is employed by methods such as TREND (Wen & Fang, 2022) and LDG (Knyazev et al., 2021), etc. Drawing inspiration from natural language processing (NLP) studies, SimpleDyG (Wu et al., 2024) models dynamic graphs as a sequence modeling problem, using a simple Transformer architecture without complex modifications. Poursafaei et al. (2022) observe that edges reoccur over time in the existing datasets and propose a simple memory-based heuristic approach, EdgeBank, without any learnable components. This method predicts edges based solely on past observations, yet it demonstrated remarkable performance in current evaluations. This further highlights the need for more comprehensive benchmarks that assess models' ability to generalize to unseen edges, thereby ensuring robust performance in real-world scenarios.

## 3 TASK FORMULATION AND CURRENT PITFALLS

Temporal graphs represent entities in the dynamic systems as nodes and interactions among entities as edges. Each edge is labeled with a timestamp to indicate the time of interaction occurred. Existing studies mainly categorize temporal graphs into two types: continuous-time temporal graphs and discrete-time temporal graphs. In this paper, we focus on continuous-time temporal graphs since they better reflect how dynamic graphs form incrementally in real-world scenarios and discrete-time temporal graphs can be directly converted to continuous-time temporal graphs without information loss. Formally, a continuous-time temporal graph can be denoted as $\mathcal{G} = (\mathcal{V}, \mathcal{E})$, where the edge set $\mathcal{E}$ can be represented as a stream of timestamped edges, i.e., $\mathcal{E} = \{(s_0, d_0, t_0), (s_1, d_1, t_1), \cdots, (s_T, d_T, t_T)\}$ with $s_i, d_i \in \mathcal{V}$ representing the source and destination nodes, respectively. The $t_i$ denotes the timestamp of the $i$-th edge with $t_0 \leq t_1 \leq \cdots \leq t_T$.

### 3.1 FUTURE LINK PREDICTION FORMULATION AND EVALUATION

**Future Link Prediction.** The task of future link prediction is formulated as predicting the existence of a link between two nodes at a given timestamp in literature (Kumar et al., 2019). Specifically, given a temporal graph $\mathcal{G}$, a query edge $(s, d, t)$, and all edges appeared before time $t$, the model is required to predict the likelihood of the edge $(s, d)$ appearing at time $t$. However, in real-world applications, the fundamental objective is to determine which entities the query entity is most likely to interact with. For instance, in the "Who-To-Follow" scenario within social networks, the task is to predict which users the query user is likely to follow next. The users with the highest predicted likelihood are then recommended to the query user. Given the high computational costs associated with calculating the likelihood of all potential entities in a large-scale graph, current literature in recommendation and knowledge graphs He et al. (2017); Kang & McAuley (2018); Teru et al. (2020) treats the future link prediction task as a ranking problem among multiple negative samples. Specifically, given a query edge $(s, d, t)$, the model needs to rank the positive destination node $d$ higher among the sampled $k$ negative destinations based on the likelihood. The current temporal graph benchmark study, TGB (Huang et al., 2024b), adopts these settings and sets $k$ to 20. We set $k$ to 100 for a more robust evaluation in our experimental setup.

**Negative Sampling Strategies.** Previous studies (Poursafaei et al., 2022; Huang et al., 2024b) leverage historical edges as negative samples to increase the difficulty for models in predicting potential links, based on the assumption that positive edges are likely repetitions of historical edges. However, this assumption does not hold in the domains covered by our TGB-Seq datasets, where historical edges are unlikely to reoccur in future time steps. Consequently, we randomly sample negative destination nodes from all possible nodes, specifically selecting from all nodes in non-bipartite datasets and all items in bipartite recommendation datasets.

**Evaluation Metrics.** Most existing studies leverage Area Under the Receiver Operating Characteristic curve (AUROC) and Average Precision (AP) for link prediction evaluation with a single negative sample, while Yang et al. (2015) and Huang et al. (2024b) argue that they are not proper metrics for

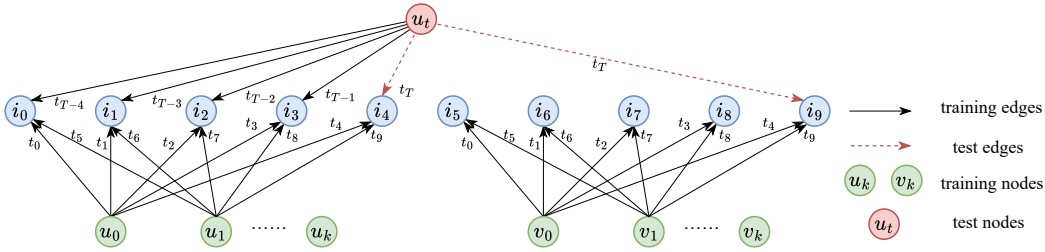

Figure 3: Toy example of sequential dynamics in a temporal graph. The bipartite graph consists of users and items. The first user in group $u$, $u_0$, interacts sequentially with items $\{i_k\}_{k=0}^{k=4}$ at time $\{t_k\}_{k=0}^{k=4}$, respectively. Similarly, the first user in group $v$, $v_0$, interacts sequentially with items $\{i_k\}_{k=5}^{k=9}$ at the same timestamps $\{t_k\}_{k=0}^{k=4}$ as $u_0$. The second users, $u_1$ and $v_1$, follow a similar interaction pattern but interact with items at different times compared to the first users. All other users interact with items in a comparable sequential manner. A test sample queries whether the test node will interact with $i_4$ or $i_9$ at time $t_T$, based on its four previous interactions from $t_{T-4}$ to $t_{T-1}$.

link prediction with multiple negative samples. Thus, we deploy the widely used ranking metric, Mean Reciprocal Rank (MRR) for evaluations, following (Cong et al., 2023; Huang et al., 2024b).

## 3.2 CURRENT PITFALLS IN TEMPORAL GNNS

In this section, we aim to demonstrate that existing temporal GNNs are unable to capture even simple sequential dynamics. Figure 3 illustrates a toy example of sequential dynamics in a temporal graph. To empirically evaluate whether existing temporal GNNs can learn the simple sequential dynamics, we construct a dataset that mirrors the dynamics depicted in Figure 3. Specifically, the dataset consists of items $\{i_k\}_{k=0}^{k=9}$ and multiple nodes in both group $u$ and group $v$, as in the toy example. To ensure that the sequential dynamics can be effectively modeled, the number of nodes in both group $u$ and group $v$ is set to 500. Each $u_k$ interacts sequentially with items $\{i_k\}_{k=0}^{k=4}$, while each $v_k$ interacts sequentially with items $\{i_k\}_{k=5}^{k=9}$. Note that each $u_k$ and $v_k$ always interact at the same timestamps as stated in the caption of Figure 3. Both nodes and edges lack features. The dataset is chronologically split into training set, validation set, and test set. The training set contains the complete interactions of 70% of the users in both group $u$ and group $v$. Given the four historical interactions, i.e, $\{i_k\}_{k=0}^{k=3}$ or $\{i_k\}_{k=5}^{k=8}$, a temporal GNN model is required to predict the interaction likelihood of the query user with $i_4$ and $i_9$. Despite these straightforward sequential dynamics appearing commonly in the training set and thus considered as simple patterns, existing methods cannot correctly predict item $i_4$ instead of $i_9$ given that a test node has interacted with $\{i_k\}_{k=0}^{k=3}$ sequentially. We use the AP metric to evaluate nine temporal GNNs and SGNN-HN. All temporal GNNs achieve an AP score of approximately 50% as shown in Table 1, indicating that they cannot distinguish between $i_4$ and $i_9$.

Table 1: The AP metric on the toy example dataset. $\ell$ indicates the length of the temporal random walk of CAWN.

| Method | AP (%) |
|---|---|
| JODIE | 51.19 ± 0.32 |
| DyRep | 51.30 ± 0.27 |
| TGAT | 51.06 ± 0.23 |
| TGN | 51.25 ± 0.48 |
| CAWN ($\ell = 1$) | 50.00 ± 0.00 |
| CAWN ($\ell = 2$) | 52.80 ± 0.05 |
| EdgeBank | 50.00 ± 0.00 |
| TCL | 50.00 ± 0.00 |
| GraphMixer | 50.00 ± 0.00 |
| DyGFormer | 50.66 ± 0.50 |
| SGNN-HN | 100.00 ± 0.00 |

The shortcomings of current temporal GNNs in capturing sequential dynamics might relate to the functionality of their structures. Generally, the temporal GNN models can be partitioned into two components: i) a memory module to represent the interaction history of the nodes, and ii) an aggregation module to aggregate neighborhood information when predicting future interactions. Among the existing studies, the designed temporal GNN models might contain both or either of these two components. The limitations of each component in capturing sequential dynamics to distinguish items $i_4$ and $i_9$ are discussed as follows.

**Notations.** We denote the node feature of $u$ as $\mathbf{x}_u$, the edge feature of $(u, v)$ as $\mathbf{e}_{u,v}$, and the time interval between the interaction $(u, v)$ and the query time as $\Delta t$. $\mathbf{mem}(u)$ and $\mathbf{emb}(u)$ denote the memory and embedding of node $u$, respectively. $\mathcal{N}_t^k(u)$ denotes the set of $k$-hop historical neighbors of node $u$ before time $t$. We use $\mathcal{N}_\mathbf{b}(u)$ to denote the set of $u$'s neighbors within a batch.

**Memory module.** The memory module is designed to memorize the interaction history of nodes using a low-dimensional representation, called *memory*. Formally, a node $s$'s memory is updated when processing a batch of incoming edges that involves $s$:

$$\mathbf{mem}(s) = \text{MEM}\left(\mathbf{mem}(s), \mathbf{x}_s, \{(\mathbf{mem}(d), \mathbf{e}_{s,d}, \Delta t) \mid d \in \mathcal{N}_{\mathbf{b}}(s)\}\right), \tag{1}$$

where MEM is typically an RNN model such as LSTM and GRU (Cho et al., 2014; Graves & Graves, 2012). The memory and feature of $s$, $\mathbf{mem}(s)$ and $\mathbf{x}_s$, are treated as the hidden state of the RNN. The information of incoming edges in the batch, $\{(\mathbf{mem}(d), \mathbf{e}_{s,d}, \Delta t) \mid d \in \mathcal{N}_{\mathbf{b}}(s)\}$, serve as the input to RNN. Typically, only the last edge for each node in the batch is considered. In the toy example, items $i_4$ and $i_9$ always interact at the same timestamps, resulting in undistinguishable memories.

**Aggregation module with one-hop historical neighbors.** Given a prediction request for potential edges $(s, d, t)$, the aggregation module aggregates the information from the historical neighbors of node $s$ and $d$ before time $t$ to generate their current embeddings. The aggregation operation for node $s$ could be formulated as:

$$\mathbf{emb}(s) = \text{AGGR}\left(\mathbf{x}_s, \{(\mathbf{x}_d, \mathbf{e}_{s,d}, \Delta t) \mid d \in \mathcal{N}_t^k(s)\}\right), \tag{2}$$

where AGGR is commonly implemented as a Transformer (Vaswani, 2017) (e.g., in DyGFormer) or its variants, an MLP-mixer (Tolstikhin et al., 2021) (e.g., in GraphMixer), or time projection function (e.g., in JODIE). Note that if the memory is available, $\mathbf{x}_s$ is replaced by a combination of the memory and node feature. In addition to interaction information, several studies compute the correlations between the neighborhoods of $s$ and $d$ to capture their structural and temporal dependencies:

$$\mathbf{rel}(s, d) = \text{CO-REL}\left(\mathcal{N}_t^k(s), \mathcal{N}_t^k(d)\right), \tag{3}$$

where CO-REL is the correlation function, such as calcu the number of common neighbors in DyG-Former and anonymous temporal random walk in CAWN. For computational efficiency, aggregation modules typically consider only one-hop neighbors.

Such aggregation modules fail to capture the sequential dynamics in the toy example. The underlying issue is similar to that of memory modules: a node is represented solely by its null features and interaction timestamps. However, $i_4$ and $i_9$, their one-hop neighbors $\{u_k\}$ and $\{v_k\}$, interact in a similar manner at identical timestamps, respectively. As a result, the aggregation modules generate the same embeddings for $i_4$ and $i_9$, as well as for $\{u_k\}$ and $\{v_k\}$, respectively. While computing correlations between the source and destination nodes may seem helpful, both DyGFormer and CAWN fail in the toy example. DyGFormer's CO-REL is ineffective when the source and destination nodes have no common neighbors. Though CAWN's CO-REL employ a sophisticated anonymous random walk technique, it fails to distinguish between $i_4$ and $i_9$ because their one-hop neighborhoods mirror each other. Therefore, the aggregation modules of existing temporal GNNs cannot capture even the simple sequential dynamics in the toy example.

**Aggregation module with high-order historical neighbors.** Leveraging high-order historical neighbor information can modestly enhance the capture of sequential dynamics. For example, extending the length of the temporal random walk from 1-hop to 2-hop in CAWN enables the incorporation of higher-order temporal and structural entangled information, resulting in a slight performance improvement from 50.00% to 52.80%. The limited gain arises because an increased number of high-order neighbors introduces excessive noise. Consequently, CAWN is unable to effectively differentiate the subtle differences between the local structures of $(u_k, i_4)$ and $(u_k, i_9)$. Furthermore, utilizing high-order information results in substantial computational resource consumption (Besta et al., 2024). CAWN encounters memory issues on a GPU with 80GB of memory when the walk length is extended to three, even on this small graph. Therefore, effectively capturing intricate sequential dynamics through high-order neighbors remains an open problem.

In summary, neither the memory module nor the aggregation module can distinguish items $i_4$ and $i_9$ in the toy example. Consequently, temporal GNNs that incorporate either or both of these modules are unable to effectively capture the simple sequential dynamics, resulting in suboptimal performance on the toy dataset. These findings suggest that current methods are insufficient for future link prediction that involve complex sequential dynamics and highlight the urgent need to establish new benchmark datasets to challenge temporal GNNs with sequential dynamics.

Table 2: Statistics of TGB-Seq datasets.

| Dataset | Nodes (users/items) | Edges | Timestamps | Repeat ratio(%) | Density(%) | Bipartite | Domain |
|---------|---------------------|-------|-----------|-----------------|-----------|-----------|--------|
| ML-20M | 100,785/9,646 | 14,494,325 | 9,993,250 | 0 | $1.49 \times 10^{0}$ | ✓ | Movie rating |
| Taobao | 760,617/863,016 | 18,853,792 | 139,171 | 16.58 | $2.87 \times 10^{-3}$ | ✓ | E-commerce interaction |
| Yelp | 1,338,688/405,081 | 19,760,293 | 14,646,734 | 25.18 | $3.64 \times 10^{-3}$ | ✓ | Business review |
| GoogleLocal | 206,244/267,336 | 1,913,967 | 1,771,060 | 0 | $3.47 \times 10^{-3}$ | ✓ | Business review |
| Flickr | 233,836 | 7,223,559 | 134 | 0 | $1.32 \times 10^{-2}$ | ✗ | Who-To-Follow |
| YouTube | 402,422 | 3,288,028 | 203 | 0 | $2.03 \times 10^{-3}$ | ✗ | Who-To-Follow |
| Patent | 2,241,784 | 12,749,824 | 1,632 | 0 | $2.54 \times 10^{-4}$ | ✗ | Citation |
| WikiLink | 1,361,972 | 34,163,774 | 2,198 | 0 | $1.84 \times 10^{-3}$ | ✗ | Web link |

## 4 PROPOSED DATASETS

Our proposed TGB-Seq aims to challenge temporal GNNs with intricate sequential dynamics that are inherently exhibited in various real-world dynamic systems. TGB-Seq comprises eight temporal graph datasets, including four bipartite datasets derived from recommender systems and four non-bipartite datasets curated from diverse application domains. All TGB-Seq datasets focus on interactions between entities and exclude node and edge features. Table 2 presents the statistics of TGB-Seq datasets. Besides, we also provide a selected list of datasets used for continuous-time temporal graph learning in Table 5 for comparison.

The most distinguishable feature of TGB-Seq datasets is the *low repeat ratio*, where only the Yelp and Taobao datasets contain repeated edges due to the natural behavior of users who may click on items multiple times. The repeat ratio $r$ is defined as the portion of the number of repeated edges to the total number of edges in the dataset, i.e., $r = \frac{|\mathcal{E}_{\text{seen}}|}{|\mathcal{E}|}$, where an edge $e_i = (s_i, d_i, t_i) \in \mathcal{E}_{\text{seen}}$ if there exists an edge $e_j = (s_j, d_j, t_j)$ and satisfies that $s_i = s_j, d_i = d_j, t_j < t_i$.

*Remark.* The phenomenon of existing datasets that contain excessive repeated edges and its impact on overly optimistic evaluations has been highlighted in previous studies. To address the issues, these studies challenge the existing temporal GNNs with new evaluation protocols and new datasets from diverse domains. Specifically, Poursafaei et al. (2022) proposes a historical negative sampling strategy to challenge existing methods with hard negative samples, and Huang et al. (2024b) further employs multiple negative sampling strategies. Both of them propose new datasets from diverse domains and of diverse scales. However, most of the proposed datasets still contain numerous repeated edges as shown in Table 5. In contrast, we address this issue by proposing new challenging datasets curated to minimize repeated edges. Our TGB-Seq datasets emphasize intricate sequential dynamics, a key characteristic of many real-world applications. Consequently, TGB-Seq datasets provide a robust benchmark for evaluating the ability of temporal GNNs to capture sequential dynamics and generalize to unseen edges, a capability that is often lacking in existing benchmark datasets.

In addition to the low repeat ratio, another notable feature of TGB-Seq datasets is their origin in *diverse domains that represent typical real-world applications of future link prediction*. Besides classical applications like recommendations, the proposed non-bipartite datasets also represent fundamental applications in real-world contexts. As a crucial task for online social networking platforms, "Who-To-Follow" aims to recommend a list of users that a given user may be interested in following (Gupta et al., 2013). Effective prediction of relevant connections between users can significantly enhance user experience by fostering engagement and interaction within the platform. Moreover, future link prediction in citation networks and web link networks can be applied to knowledge graph completion, thereby enriching knowledge representations and enabling more comprehensive information retrieval (Wang et al., 2021b). Furthermore, TGB-Seq datasets exhibit several key attributes of real-world networks. Specifically, all TGB-Seq datasets adhere to a *power-law degree distribution* (see Appendix B, Figure 5) and are notably sparse. Additionally, each dataset is *medium to large scale*, containing millions to tens of millions of edges, which is consistent with the scale typically encountered in real-world networks.

**Dataset preprocessing.** We split the datasets chronologically into training, validation, and test sets with a ratio of 70%/15%/15%. In the training sets, we retain only nodes with a degree of at least three. Besides, only nodes appearing in the training set are included in the validation and test sets. These settings are designed to mitigate the effects of cold-start nodes and the high sparsity of the datasets on the evaluation. Descriptions of TGB-Seq datasets are shown below.

ML-20M [2] is a widely used benchmark dataset in recommendation research, derived from the MovieLens website. It contains movie rating data, where each record includes the rating score of a user, ranging from 1 to 5, for a specific movie along with the timestamp of the rating. While the ratings represent explicit feedback, we transform this data into implicit feedback for our analysis, following He et al. (2017). Consequently, the ML-20M network is represented as a bipartite graph where users and movies serve as nodes, and an edge represents a user's rating of a movie at a given time. The task of ML-20M and the following recommendation datasets is to predict whether a given user will interact with a given item at a given time.

Taobao [3] (Zhu et al., 2018; 2019; Zhuo et al., 2020) is a user behavior dataset derived from the e-commerce platform Taobao. It contains user click data on products from November 25, 2017, to December 3, 2017. The dataset is a bipartite graph where users and products are nodes, and an edge represents a user's click on a product at a given time.

Yelp [4] is a business review dataset sourced from Yelp, a prominent platform for business recommendations, including restaurants, bars, and beauty salons. It contains user reviews of businesses from 2018 to 2022. The dataset is a bipartite graph where users and businesses are nodes, and an edge represents a user's review of a business at a given time.

GoogleLocal (Li et al., 2022; Yan et al., 2023) is a business review dataset derived from Google Maps, with a smaller scale compared to Yelp. It contains user reviews and ratings of local businesses. Following the settings for the ML-20M dataset, we treat these ratings as implicit feedback. Similar to the Yelp dataset, the GoogleLocal dataset is a bipartite graph where users and businesses are nodes, and an edge indicates a user's review of a business at a given time.

Flickr (Cha et al., 2009) is a "Who-To-Follow" social network dataset derived from Flickr, a photo-sharing platform with social networking features. The dataset was crawled daily from November 2 to December 3, 2006, and from February 3 to March 18, 2007 by  Cha et al. (2009). It is estimated to represent 25% of the entire Flickr network. The Flickr dataset is a non-bipartite graph where users are nodes, and an edge represents the friendship established between users at a given time. The task for the "Who-To-Follow" datasets, including Flickr and YouTube, is to predict whether a given user will follow another specified user at a particular time.

YouTube (Mislove et al., 2007) is another "Who-To-Follow" social network dataset derived from YouTube, a video-sharing platform that includes a user subscription network. Similar to Flickr, the YouTube dataset is a non-bipartite graph where users are nodes, and an edge indicates the subscription of a user to another user at a given time.

Patent (Hall et al., 2001) is a citation network dataset of U.S. patents, capturing the citation relationships between patents from 1963 to 1999. The dataset is organized as a non-bipartite graph where patents are nodes, and an edge represents a citation made by one patent to another at the time of publication. The task for the Patent dataset is to predict whether a given patent will cite another given patent, given several of their established citations.

WikiLink (Boldi et al., 2004; 2011; Boldi & Vigna, 2004) is a web link network dataset derived from Wikipedia, containing the hyperlink relationships between Wikipedia pages. This dataset is a non-bipartite graph, where pages are nodes and edges indicate hyperlinks established from one page to another at a given time. The task for WikiLink is to predict whether a given page will link to another given page at a given time.

## 5  EXPERIMENTS

In this section, we evaluate the performance of existing temporal GNNs on TGB-Seq datasets. The selected temporal GNN models includes JODIE (Kumar et al., 2019), DyRep (Trivedi et al., 2019), TGAT (Xu et al., 2020), TGN (Rossi et al., 2020), CAWN (Wang et al., 2021d), TCL (Wang et al., 2021a), GraphMixer (Cong et al., 2023), and DyGFormer (Yu et al., 2023). The descriptions of these methods are provided in Appendix E. We employ the DyGLib Yu et al. (2023) framework to conduct the experiments. We limit the running time of each method to 48 hours and omit the methods that

---

[2]https://grouplens.org/datasets/movielens/20m/

[3]https://tianchi.aliyun.com/dataset/dataDetail?dataId=649

[4]https://www.yelp.com/dataset

Table 3: MRR of eight popular temporal GNN methods and SGNN-HN on four recommendation datasets and two previously established datasets (e.g., Wikipedia and Reddit). "OOT" denotes that the method failed to complete one epoch of training within 24 hours. The first, second, and third place rankings are highlighted accordingly.

| Datasets | ML-20M | Taobao | Yelp | GoogleLocal | Wikipedia | Reddit |
|---|---|---|---|---|---|---|
| JODIE | 21.16 ± 0.73 | 48.36 ± 2.18 | 69.88 ± 0.31 | 41.86 ± 1.49 | 76.94 ± 0.28 | 77.92 ± 0.10 |
| DyRep | 19.00 ± 1.69 | 40.03 ± 2.40 | 57.69 ± 1.05 | 37.73 ± 1.34 | 68.09 ± 1.45 | 75.30 ± 0.30 |
| TGAT | 10.47 ± 0.20 | OOT | OOT | 19.78 ± 0.24 | 72.42 ± 0.38 | 76.69 ± 0.52 |
| TGN | 23.99 ± 0.20 | 60.28 ± 0.54 | 69.79 ± 0.24 | 54.13 ± 1.97 | 81.16 ± 0.19 | 79.82 ± 0.26 |
| CAWN | 12.31 ± 0.02 | OOT | 25.71 ± 0.09 | 18.26 ± 0.02 | 88.23 ± 0.33 | 87.31 ± 0.32 |
| TCL | 12.04 ± 0.02 | 31.55 ± 0.03 | 24.39 ± 0.67 | 18.30 ± 0.02 | 45.47 ± 3.48 | 36.09 ± 2.10 |
| GraphMixer | 21.97 ± 0.17 | 31.54 ± 0.02 | 33.96 ± 0.19 | 21.31 ± 0.14 | 72.14 ± 0.80 | 71.73 ± 0.32 |
| DyGFormer | OOT | OOT | 21.68 ± 0.20 | 18.39 ± 0.02 | 84.64 ± 0.43 | 83.57 ± 1.42 |
| SGNN-HN | 33.12 ± 0.01 | 68.58 ± 0.21 | 69.34 ± 0.44 | 62.88 ± 0.51 | 83.83 ± 0.55 | 89.01 ± 0.17 |

require more than 24 hours to finish one training epoch, which are denoted as OOT (out of time). Each result is the average of three runs with different random seeds with reported standard deviation.

**Implementation details.** We follow Rossi et al. (2020) to set a relatively small batch size to ensure timely updates for the memory module. Specifically, we set the batch size to 200 for the GoogleLocal dataset across all methods. For larger datasets, however, a batch size of 200 is too small and would incur unacceptable training costs for most methods. Thus, we increase the batch size to 400 for all other datasets to accelerate the training process. Following DyGFormer, we use a learning rate of 0.0001 across all methods and datasets. A grid search is performed to tune the hyper-parameters of each method on the validation set. Detailed configurations are provided in Appendix D.1.

## 5.1 FUTURE LINK PREDICTION PERFORMANCE

**Performance on recommendation datasets.** Table 3 presents the results on four recommendation datasets, ML-20M, Taobao, Yelp, and GoogleLocal. Additionally, the results on two existing datasets, Wikipedia and Reddit, are included for comparison. We observe that all existing temporal GNNs underperform on ML-20M, GoogleLocal and Taobao, with a large margin compared to SGNN-HN, one of the state-of-the-art methods for sequential recommendation. Notably, on the Yelp dataset, JODIE and TGN perform similarly to SGNN-HN and significantly outperform other temporal GNNs. This contrasts with the results on Wikipedia and Reddit, where memory-based models generally perform worse than CAWN and DyGFormer. In contrast, CAWN and DyGFormer do not achieve satisfactory performance across all recommendation datasets.

These observations suggest that different temporal GNNs exhibit varying abilities in capturing sequential dynamics across datasets. Popular state-of-the-art methods like CAWN and DyGFormer tend to struggle with generalizing to unseen edges in real-world applications, excelling instead in capturing repetitive patterns. Memory-based methods, particularly JODIE and TGN, on the other hand, tend to perform better on datasets with fewer repeated edges. These findings highlight the need for more versatile methods capable of addressing the diverse challenges posed by different datasets.

**Performance on non-bipartite datasets.** Table 4 presents the results on four non-bipartite datasets, Flickr, YouTube, Patent, and WikiLink. The rankings of existing temporal GNNs vary even more significantly across these datasets. DyGFormer and GraphMixer outperform other methods on Flickr and YouTube, respectively, while JODIE and TGN perform best on Patent and WikiLink. Almost all methods rank in the top three in at least one dataset, which contrasts with the performance on recommendation datasets, where only JODIE and TGN perform well. These results highlight the diverse capabilities of existing temporal GNNs and emphasize the need for more adaptable methods that can generalize across a wide range of datasets. The varying results presented in Table 3 and Table 4 underscore that the TGB-Seq datasets offer a more comprehensive evaluation of temporal GNNs' capabilities and introduce new challenges for future link prediction tasks.

## 5.2 TRAINING COST

To comprehensively study the efficiency of existing temporal GNNs, we select three datasets with various sizes of edge sets and report the average training cost per epoch of the corresponding approach.

Table 4: MRR score of eight popular temporal GNNs on four non-bipartite datasets. The first, second, and third place rankings are highlighted accordingly.

| Datasets | Flickr | YouTube | Patent | WikiLink |
|---|---|---|---|---|
| JODIE | 46.21 ± 0.83 | 41.67 ± 2.86 | 24.60 ± 0.38 | 57.94 ± 1.33 |
| DyRep | 38.04 ± 4.19 | 35.12 ± 4.13 | 21.01 ± 1.14 | 42.63 ± 1.33 |
| TGAT | 23.53 ± 3.35 | 43.56 ± 2.53 | 8.49 ± 0.18 | OOT |
| TGN | 46.03 ± 6.78 | 55.16 ± 5.89 | 22.83 ± 2.25 | 62.94 ± 2.16 |
| CAWN | 48.69 ± 6.08 | 47.55 ± 1.08 | 12.34 ± 0.47 | OOT |
| TCL | 40.00 ± 1.76 | 50.17 ± 1.98 | 10.60 ± 1.75 | 43.02 ± 2.16 |
| GraphMixer | 45.01 ± 0.08 | 58.87 ± 0.12 | 18.97 ± 2.54 | 48.57 ± 0.02 |
| DyGFormer | 49.58 ± 2.87 | 46.08 ± 3.44 | 14.20 ± 2.93 | OOT |

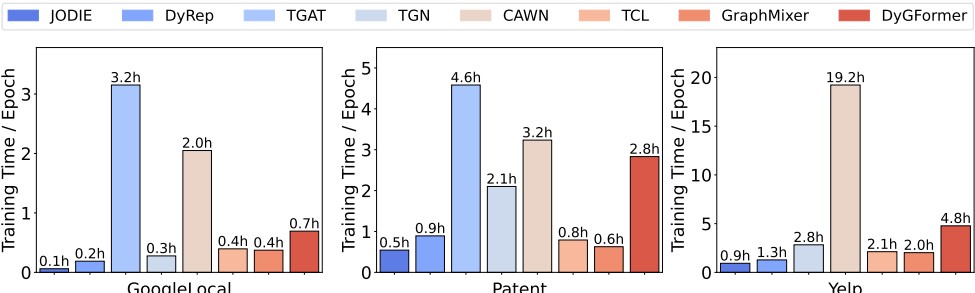

Figure 4: The average training cost per epoch of eight popular temporal GNN methods on GoogleLocal, Patent, and Yelp datasets consists of 1.9M, 12.7M, and 19.7M edges, respectively.

Figure 4 illustrates the results on the GoogleLocal, Patent, and Yelp datasets, where the methods that cannot finish one epoch in 24 hours are omitted. We observe that methods with simpler architectures, such as JODIE, DyRep, TCL and GraphMixer, exhibit significantly shorter training time compared to the others. In contrast, TGAT and CAWN are the most inefficient methods, requiring considerable time to complete an epoch on the Patent and Yelp datasets. This inefficiency stems from the complex aggregation modules of these methods, which require multi-hop neighbor retrieval. While DyGFormer is more efficient than TGAT and CAWN on GoogleLocal, Patent, and Yelp, it still incurs higher costs. The calculation of co-occurrence frequencies for neighbors becomes particularly expensive when the temporal graph is dense. As demonstrated in Table 3, DyGFormer cannot complete an epoch within 24 hours for the ML-20M dataset, whereas other methods, including TGAT and CAWN, can.

These observations highlight that complex aggregation modules can significantly increase the training cost of existing temporal GNNs. As demonstrated in Table 3 and Table 4, simpler methods like TCL and GraphMixer may be more efficient in terms of training, but they fail to achieve satisfactory performance. This investigation suggests that achieving both efficiency and effectiveness in temporal GNNs simultaneously remains an open problem, further underscoring the distinctive capability of TGB-Seq for comprehensive evaluations of these models.

## 6   CONCLUSION

In this paper, we demonstrate that current temporal GNNs struggle to capture intricate sequential dynamics inherent in real-world dynamic systems, thereby limiting their abilities to generalize across various future link prediction applications. However, existing datasets often contain excessively repeated edges and thus are inadequate for evaluating such abilities of temporal GNNs comprehensively. To address this gap, we propose TGB-Seq, a new challenging benchmark for temporal GNNs. TGB-Seq comprises eight datasets curated from diverse application domains characterized by complex sequential dynamics. Comprehensive evaluations on TGB-Seq reveal that existing temporal GNNs fail to achieve satisfactory performance across all datasets, underscoring the limitations of current methods and the necessity of TGB-Seq for robust temporal GNN evaluation.

ACKNOWLEDGEMENTS

The work was partially done at Gaoling School of Artificial Intelligence, Beijing Key Laboratory of Big Data Management and Analysis Methods, MOE Key Lab of Data Engineering and Knowledge Engineering, and Pazhou Laboratory (Huangpu), Guangzhou, Guangdong 510555, China. This research was supported in part by National Natural Science Foundation of China (No. 92470128, No. U2241212, No. 62276066), by National Science and Technology Major Project (2022ZD0114802), by Beijing Outstanding Young Scientist Program No.BJJWZYJH012019100020098, by Huawei-Renmin University joint program on Information Retrieval. We also wish to acknowledge the support provided by the fund for building world-class universities (disciplines) of Renmin University of China, by Engineering Research Center of Next-Generation Intelligent Search and Recommendation, Ministry of Education, by Intelligent Social Governance Interdisciplinary Platform, Major Innovation & Planning Interdisciplinary Platform for the "Double-First Class" Initiative, Public Policy and Decision-making Research Lab, and Public Computing Cloud, Renmin University of China.

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

Table 5: A selected list of datasets used for continuous-time temporal graph learning.

| Dataset | Nodes (users/items) | Edges | Timestamps | Repeat ratio(%) | Density(%) | Bipartite | Domain |
|---|---|---|---|---|---|---|---|
| ML-20M | 100,785/9,646 | 14,494,325 | 9,993,250 | 0 | $1.49 \times 10^0$ | ✓ | Movie rating |
| Taobao | 760,617/863,016 | 18,853,792 | 139,171 | 16.58 | $2.87 \times 10^{-3}$ | ✓ | E-commerce interaction |
| Yelp | 1,338,688/405,081 | 19,760,293 | 14,646,734 | 25.18 | $3.64 \times 10^{-3}$ | ✓ | Business review |
| GoogleLocal | 206,244/267,336 | 1,913,967 | 1,771,060 | 0 | $3.47 \times 10^{-3}$ | ✓ | Business review |
| Flickr | 233,836 | 7,223,559 | 134 | 0 | $1.32 \times 10^{-2}$ | × | Who-To-Follow |
| YouTube | 402,422 | 3,288,028 | 203 | 0 | $2.03 \times 10^{-3}$ | × | Who-To-Follow |
| Patent | 2,241,784 | 12,749,824 | 1,632 | 0 | $2.54 \times 10^{-4}$ | × | Citation |
| WikiLink | 1,361,972 | 34,163,774 | 2,198 | 0 | $1.84 \times 10^{-3}$ | × | Web link |
| Wikipedia | 8,227/1,000 | 157,474 | 152,757 | 88.41 | $1.91 \times 10^0$ | ✓ | Interaction |
| Reddit | 10,000/984 | 672,447 | 669,065 | 88.32 | $6.83 \times 10^0$ | ✓ | Social |
| MOOC | 7,047/97 | 411,749 | 345,600 | 56.66 | $6.02 \times 10^1$ | ✓ | Interaction |
| LastFM | 980/1,000 | 1,293,103 | 1,283,614 | 88.01 | $1.32 \times 10^2$ | ✓ | Interaction |
| Enron | 184 | 125,235 | 22,632 | 90.79 | $3.70 \times 10^2$ | × | Social |
| Social Evo. | 74 | 2,099,519 | 565,932 | 99.77 | $3.83 \times 10^4$ | × | Proximity |
| UCI | 1,899 | 59,835 | 58,911 | 66.06 | $1.66 \times 10^0$ | × | Social |
| Flights | 13,169 | 1,927,145 | 122 | 79.50 | $1.11 \times 10^0$ | × | Transport |
| Contact | 692 | 2,426,279 | 8,064 | 96.72 | $5.07 \times 10^2$ | × | Proximity |
| tgbl-wiki | 8,227/1,000 | 157,474 | 152,757 | 88.41 | $1.91 \times 10^0$ | ✓ | Interaction |
| tgbl-review | 352,636/298,590 | 4,873,540 | 6,865 | 0.19 | $4.63 \times 10^{-3}$ | ✓ | Rating |
| tgbl-coin | 638,486 | 22,809,486 | 1,295,720 | 82.93 | $5.60 \times 10^{-3}$ | × | Transaction |
| tgbl-comment | 994,790 | 44,314,507 | 30,998,030 | 19.81 | $4.48 \times 10^{-3}$ | × | Social |
| tgbl-flight | 18,143 | 67,169,570 | 1,385 | 96.48 | $2.04 \times 10^1$ | × | Transport |
| Bitcoin-Alpha | 3,783 | 24,186 | 24,186 | 0 | $1.69 \times 10^{-1}$ | × | Finance |
| Bitcoin-OTC | 5,881 | 35,592 | 35,592 | 0 | $1.03 \times 10^{-1}$ | × | Finance |

## A  DATASET DOCUMENTATION

The resource links for TGB-Seq benchmark suits and datasets are provided as follows.

- The TGB-Seq website: `https://tgb-seq.github.io/`.

- The TGB-Seq datasets are available at Hugging Face: `https://huggingface.co/TGB-Seq`.

- The `tgb-seq` pip package is at `https://pypi.org/project/tgb-seq/`. The package will download datasets and negative samples automatically from Hugging Face.

- The TGB-Seq datasets can also be downloaded at `https://drive.google.com/drive/folders/1qoGtASTbYCO-bSWAzSqbSY2YgHr9hUhK`.

- The original datasets of TGB-Seq are available at `https://drive.google.com/file/d/1-1Ndp3R2qk_Jfk2zctReZIPOOzUuAQQI`.

## B  FURTHER INFORMATION ON TGB-SEQ DATASETS

We provide a selected list of commonly used datasets for continuous-time temporal graph learning in Table 5 for reference. We also plot the node degree distribution of TGB-Seq datasets in Figure 5. Figure 5 demonstrates that all the TGB-Seq datasets *follow a power-law distribution*, a common characteristic of real-world networks (Barabási, 2013). This power-law behavior indicates that while a few hub nodes have a high degree of connections, the majority of nodes possess significantly fewer connections. Consequently, TGB-Seq is *highly sparse*, exhibiting low density, as shown in Table 5.

**Preprocessing of the Patent dataset.** The Patent dataset is quite special that all citations of one patent are labeled with the same timestamp, specifically the publication time of the patent. Therefore, we carefully select test samples to ensure that each patent has prior citations, so that temporal GNNs are able to leverage these historical edges for future link prediction. Specifically, we choose not to validate or test the first 50% of citations for the patents included in the validation and test sets; these citations serve solely as historical edges and are not used for model training. The remaining 50% of citations are then evenly divided into validation and test samples. Although the citations of a patent occur simultaneously at the publication time, temporal GNNs can utilize the relative publication times of these patents and their neighbors to capture inherent research trends, thereby enhancing future link prediction performance.

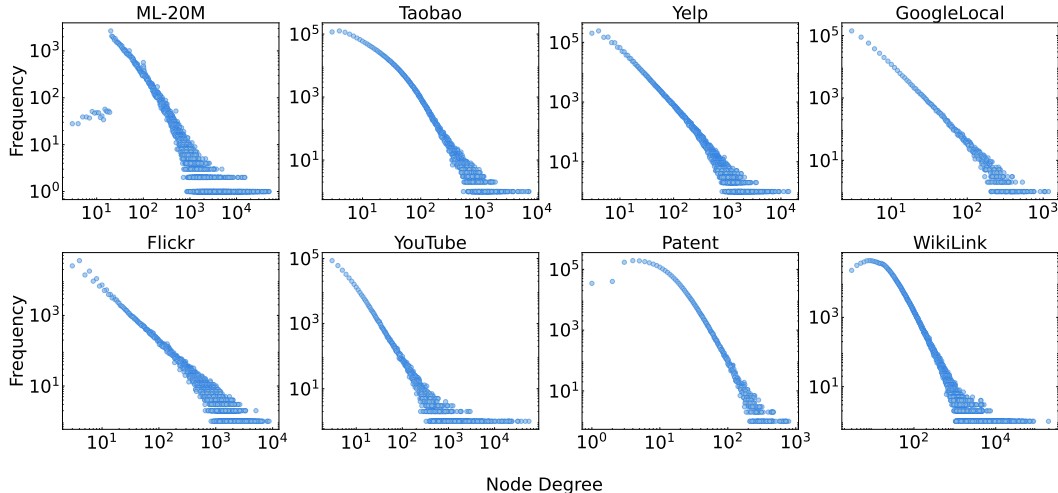

Figure 5: Distribution of node degree on our TGB-Seq dataset.

Table 6: Performance comparison across models and various feature configurations on GoogleLocal.

|              | JODIE | DyRep | TGAT  | TGN   | CAWN  | TCL   | GraphMixer | DyGFormer |
|--------------|-------|-------|-------|-------|-------|-------|------------|-----------|
| **w/o features**  | 41.86 | 37.73 | 19.78 | 54.13 | **18.26** | **18.30** | 21.31 | 18.39 |
| **+ edge features** | **42.44** | 37.22 | 17.35 | 53.48 | 15.76 | 8.58  | 21.38 | **18.53** |
| **+ node features** | 42.28 | **47.41** | **30.50** | **60.34** | 15.66 | 14.46 | **21.51** | 17.91 |

## B.1 NODE FEATURES AND EDGE FEATURES IN TGB-SEQ DATASETS

Most of TGB-Seq datasets, including all non-bipartite datasets and Taobao, do not include additional node features or edge features. However, other datasets, such as ML-20M and Yelp, include extra text features such as movies tags and businesses information. However, due to the absence of user-specific features, we cannot align the node features across users and items (i.e., movies and businesses in ML-20M and Yelp, respectively). The only dataset that includes additional text features for both items and users is GoogleLocal.

In the following, we extract the text features from GoogleLocal and perform an empirical analysis of the impact of node and edge features on the performance of temporal GNNs. Specifically, the original GoogleLocal dataset contains node features for users and places, including *name*, *jobs*, *currentPlace*, *previousPlace*, *education* for users, and *name*, *price*, *address*, *hour*, *closed* for places. We treat *price* and *closed* as one-dimensional features, while the combination of other features is processed using SBERT (Reimers & Gurevych, 2019) to generate semantic embeddings. These semantic embeddings are then reduced in dimensionality via PCA, resulting in a final embedding size of 172. Similarly, the edge features are processed as follows: the dataset includes user reviews of places, which consist of *rating*, *review_text*, and *category*. The *rating* is treated as a one-dimensional feature, while *review_text* and *category* are combined and processed using SBERT. The final dimensionality of edge features is also 172. SBERT is chosen for its ability to capture the semantic information of multilingual text, which is essential as the text in GoogleLocal is multilingual.

Table 6 demonstrates the performance of existing temporal GNNs on the GoogleLocal dataset when incorporating node features or edge features. The results indicate that the inclusion of node features significantly improves the performance of DyRep, TGAT, and TGN, while other methods show minimal improvement or even performance degradation when node or edge features are included. These findings suggest that while node features can enhance the performance of temporal GNNs, not all methods are equally effective at leveraging them for future link prediction tasks. This highlights the challenges of incorporating features into temporal GNNs and underscores the need for more robust and effective feature integration strategies in future research.

Additionally, it is important to note that the absence of features in future link prediction is a common scenario in real-world applications. A robust future link predictor must be able to capture the underlying dynamics based solely on interaction data. This is because features in real-world dynamic

graphs are often incomplete, noisy, or difficult to align across different node types, especially in bipartite and heterogeneous graphs. For example, in the GoogleLocal dataset, user features and place features cannot be aligned due to their entirely different semantic meanings. Moreover, 267,200 out of 267,336 places in GoogleLocal lack the price feature entirely, further complicating the use of node features. These practical challenges explain the absence of features or the reliance on low-dimensional features in many existing temporal graph benchmarks. On the other hand, interaction data often plays a more critical role than feature data in link prediction tasks. For instance, SGNN-HN, which does not utilize any features, achieves the best performance among all temporal GNN models on GoogleLocal, including those that incorporate features. This highlights the pivotal importance of temporal interaction data in link prediction and emphasizes the significant progress that is still required for temporal graph learning models to fully harness the potential of such data.

## B.2 DATASET LICENSES AND DOWNLOAD LINKS

In this section, we provide dataset licenses and download links as follows.

ML-20M: The data set may be used for any research purposes under the following conditions: (a) The user may not state or imply any endorsement from the University of Minnesota or the GroupLens Research Group. (b) The user must acknowledge the use of the data set in publications resulting from the use of the data set. (c) The user may not redistribute the data without separate permission. (d) The user may not use this information for any commercial or revenue-bearing purposes without first obtaining permission from a faculty member of the GroupLens Research Project at the University of Minnesota. (e) The executable software scripts are provided "as is" without warranty of any kind, either expressed or implied, including, but not limited to, the implied warranties of merchantability and fitness for a particular purpose. The entire risk as to the quality and performance of them is with you. Should the program prove defective, you assume the cost of all necessary servicing, repair or correction. (f) In no event shall the University of Minnesota, its affiliates or employees be liable to you for any damages arising out of the use or inability to use these programs (including but not limited to loss of data or data being rendered inaccurate). The original dataset can be found here.

Taobao: CC BY-NC-SA 4.0 license (Creative Commons Attribution-NonCommercial-ShareAlike 4.0 International). The original dataset can be found here.

Yelp: MIT license. The original dataset can be found here.

GoogleLocal: The original dataset can be found here.

Flickr: CC BY-SA license (Creative Commons Attribution-ShareAlike). The original dataset can be found here.

YouTube: CC BY-SA license (Creative Commons Attribution-ShareAlike). The original dataset can be found here.

Patent: MIT license. The original dataset can be found here.

WikiLink: CC BY-SA license (Creative Commons Attribution-ShareAlike). The original dataset can be found here.

## C FURTHER DISCUSSION ON REPEAT AND EXPLORATION BEHAVIORS IN TEMPORAL GRAPHS

We explore the future link prediction task in temporal graphs from the perspective of seen and unseen edges in Section 1, which are analogous to repeat and exploration behaviors in recommendation systems. Actually, repeat and exploration behaviors of users have been extensively studied in the context of recommender systems. Repeat behavior refers to users consistently engaging with items they have previously interacted with (i.e., the reoccurrence of seen edges in temporal graphs), while exploration behavior involves users discovering new items they have not interacted with before (i.e., the appearance of unseen edges for the first time in temporal graphs). Existing studies reveal an imbalance in accuracy and difficulty between repetition and exploration in sequential recommendation tasks (Li et al., 2023b). Several methods have been proposed to better address repeat and exploration behaviors, particularly in session-based or sequential recommendation (Ren et al., 2019; Chang et al.,

2024), as well as next-basket recommendation (Li et al., 2024; 2023a). However, while repeat and exploration behaviors have been extensively studied in recommendation scenarios, their conclusions may not directly apply to the future link prediction task in temporal graphs due to differences in model design and task settings. For example, many recommendation methods are tailored for bipartite graphs without features or interaction timestamps, whereas temporal GNNs often focus on general graphs that may include single or multiple node types and fully leverage temporal graph information such as features and interaction timestamps. Therefore, it is essential to investigate repeat and exploration behaviors in the context of future link prediction tasks on temporal graphs and to comprehensively evaluate the performance of existing temporal GNNs in handling these challenges. This is a key motivation behind the design of the TGB-Seq benchmark, which aims to provide a comprehensive evaluation of temporal GNNs across various datasets with diverse repeat and exploration behaviors.

## D  EXPERIMENTS DETAILS

### D.1  EXPERIMENTAL CONFIGURATIONS

We perform a grid search to determine the optimal settings for key hyperparameters, with the search ranges and corresponding methods detailed in Table 7. The final hyperparameter configurations identified through the grid search for various methods are summarized in Table 8 and Table 9. Regarding the configurations of neighbor sampling strategies, most methods achieve their best performance using the recent neighbor sampling strategy. However, for the ML-20M dataset, both CAWN and TCL achieve their best performance with the uniform neighbor sampling strategy.

For the ML-20M and the Flickr datasets, experiments are conducted on an Ubuntu machine equipped with Intel(R) Xeon(R) Gold 6240R CPU @ 2.40GHz. The GPU device is NVIDIA A100 with 80 GB memory. For the Taobao dataset, experiments are conducted on an Ubuntu machine equipped with Intel(R) Xeon(R) Gold 5218 CPU @ 2.30GHz. The GPU device is NVIDIA A100-SXM4 with 80 GB memory. For the Yelp dataset, experiments are conducted on an Ubuntu machine equipped with Intel(R) Xeon(R) Gold 6226R CPU @ 2.90GHz. The GPU device is NVIDIA RTX A6000 with 40 GB memory. For the GoogleLocal, the Patent, and the WikiLink datasets, experiments are conducted on an Ubuntu machine equipped with Hygon C86 7390 32-core Processor. The GPU device is NVIDIA A800 with 80 GB memory. For the YouTube dataset, experiments are conducted on an Ubuntu machine equipped with Intel(R) Xeon(R) Platinum 8369B CPU @ 2.90GHz. The GPU device is A100-SXM4 with 80 GB memory.

Table 7: Searched ranges of hyperparameters and the related methods.

| Hyperparameters | Searched Ranges | Related Methods |
|---|---|---|
| Number of Sampled Neighbors | [20, 30, 40, 50, 60] | DyRep, TGAT, TGN, CAWN, TCL, GraphMixer |
| Dropout Rate | [0.1, 0.3, 0.5] | JODIE, DyRep, TGAT, TGN, CAWN, TCL, GraphMixer, DyGFormer |
| Neighbor Sampling Strategies | [recent, uniform] | DyRep, TGAT, TGN, CAWN, TCL, GraphMixer |
| Length of Input Sequences & Patch Size | [32 & 1, 64 & 2] | DyGFormer |

## E  TEMPORAL GRAPH LEARNING METHODS

We provide a brief overview of the temporal GNNs used in our experiments as follows.

JODIE (Kumar et al., 2019) uses two coupled recurrent neural networks to dynamically update the states of users and items during interactions. It includes a novel projection operation that predicts future representation trajectories of both users and items, allowing the model to anticipate future behaviors. This architecture not only captures the evolution of user-item interactions but also

Table 8: Configurations of the number of sampled neighbors and the length of input sequences & the patch size of different methods.

| Datasets | DyRep | TGAT | TGN | CAWN | TCL | GraphMixer | DyGFormer |
|----------|-------|------|-----|------|-----|------------|-----------|
| ML-20M | 20 | 50 | 40 | 60 | 60 | 60 | - |
| Taobao | 20 | - | 40 | 60 | 60 | 60 | - |
| Yelp | 20 | - | 40 | - | 60 | 60 | 32 & 1 |
| GoogleLocal | 20 | 60 | 40 | 60 | 60 | 20 | 64 & 2 |
| Flickr | 60 | 40 | 20 | 40 | 50 | 40 | 32 & 1 |
| YouTube | 20 | 40 | 60 | 50 | 40 | 50 | 32 & 1 |
| Patent | 60 | 40 | 60 | 40 | 40 | 40 | 64 & 2 |
| WikiLink | 40 | - | 20 | - | 60 | 50 | - |

Table 9: Configurations of the dropout rate of different methods.

| Datasets | JODIE | DyRep | TGAT | TGN | CAWN | TCL | GraphMixer | DyGFormer |
|----------|-------|-------|------|-----|------|-----|------------|-----------|
| ML-20M | 0.1 | 0.1 | 0.3 | 0.1 | 0.1 | 0.1 | 0.3 | - |
| Taobao | 0.1 | 0.1 | - | 0.1 | 0.1 | 0.1 | 0.1 | - |
| Yelp | 0.1 | 0.1 | - | 0.1 | - | 0.1 | 0.3 | 0.1 |
| GoogleLocal | 0.1 | 0.3 | 0.1 | 0.1 | 0.1 | 0.1 | 0.1 | 0.1 |
| Flickr | 0.1 | 0.1 | 0.1 | 0.3 | 0.1 | 0.1 | 0.1 | 0.1 |
| YouTube | 0.3 | 0.1 | 0.1 | 0.1 | 0.1 | 0.1 | 0.1 | 0.1 |
| Patent | 0.3 | 0.3 | 0.1 | 0.1 | 0.1 | 0.1 | 0.1 | 0.1 |
| WikiLink | 0.1 | 0.1 | - | 0.3 | - | 0.1 | 0.1 | - |

facilitates the learning of representations that can be used for downstream tasks like recommendation and link prediction.

DyRep (Trivedi et al., 2019) introduces a dynamic representation learning framework that updates node states in real-time with each interaction. It leverages a recurrent neural network to capture node interactions and utilizes a temporal-attentive aggregation module to focus on evolving graph structures over time. DyRep is particularly effective in modeling dynamic relationships by considering both node communication and structural events, thus providing a comprehensive understanding of temporal graph changes.

TGAT (Xu et al., 2020) incorporates self-attention mechanisms to simultaneously model both the structural and temporal properties of dynamic graphs. Its design includes a time encoding function that uniquely represents temporal information, enabling the model to handle complex, evolving interactions among nodes. This combination allows TGAT to capture intricate temporal patterns and efficiently aggregate information from temporal-topological neighbors.

TGN (Rossi et al., 2020) introduces a memory-based approach for dynamic graph learning, where each node maintains an evolving memory that is updated through various interactions. Using a combination of message functions, aggregators, and memory updaters, TGN generates temporal node representations. The embedding module is crucial in capturing the temporal dynamics of nodes, which makes TGN adaptable for various dynamic graph tasks like link prediction and node classification.

CAWN (Wang et al., 2021d) performs random walks on continuous-time dynamic graphs and employs an attention mechanism to selectively focus on crucial segments of these walks. This allows it to capture both temporal relationships and causal dependencies in the network. By learning these patterns, CAWN is capable of generating relative node identities, making it effective for temporal graph tasks such as anomaly detection and node classification.

EdgeBank (Poursafaei et al., 2022) is a memory-centric approach tailored for transductive dynamic link prediction without relying on trainable parameters. It memorizes observed interactions and uses various strategies to update its memory. EdgeBank predicts future interactions based on whether the

interaction is stored in its memory. Its simplicity lies in its rule-based decision-making, making it a lightweight yet competitive approach for link prediction in dynamic networks.

TCL (Wang et al., 2021a) employs contrastive learning on temporal graphs to learn robust node embeddings. Maximizing the agreement between node pairs that are temporally similar captures both temporal dependencies and topological structures. TCL uses a graph transformer to incorporate both graph topology and temporal information, along with cross-attention mechanisms to model interactions between nodes over time.

GraphMixer (Cong et al., 2023) focuses on enhancing node embeddings in dynamic graphs by mixing both temporal and structural features. It uses a fixed time encoding function rather than a trainable one, incorporating it into a link encoder based on MLP-Mixer to learn temporal links effectively. GraphMixer also includes a node encoder with neighbor mean-pooling to aggregate node features, offering a comprehensive method for dynamic graph analysis.

DyGFormer (Yu et al., 2023) adopts a Transformer-based approach to capture long-term temporal dependencies in dynamic graphs. It introduces neighbor co-occurrence encoding and patching techniques, which help in modeling both the local and global structure of evolving interactions. This allows DyGFormer to effectively capture complex patterns in dynamic environments, making it suitable for various temporal graph tasks.

