# OpenReview forum: "TGB-Seq Benchmark: Challenging Temporal GNNs with Complex Sequential Dynamics"
_ICLR.cc/2025/Conference — ICLR 2025 Poster_

### Official Review · Reviewer_UMN2 · 2024-10-20

**Soundness:** 3
**Presentation:** 3
**Contribution:** 3
**Rating:** 6
**Confidence:** 3

**Summary:**

The paper begins with an experimental investigation exploring the comparison of recent temporal GNN methods on the recommendation datasets, where the results are surprisingly lower than the classical methods in the recommendation. The paper then investigates the reason for the datasets and proposes a more comprehensive benchmark to compare the existing methods more effectively. This paper presents an important perspective, often overlooked by many in the field, to reevaluate whether the current approach to future link prediction tasks is reasonable. I believe this work is valuable for encouraging a rethinking within the field and can inspire future research efforts.

**Strengths:**

1. This paper introduces a significant perspective that many in the field have often overlooked. It prompts a reevaluation of whether the current methods used for future link prediction tasks are truly appropriate.
2. This paper conducts insightful data analysis on existing datasets to examine the current evaluation standards.

**Weaknesses:**

1. In the illustration of Figure 3, the node features are omitted. However, this omission is not appropriate given that incorporating node features is one of the key advantages of Graph Neural Networks (GNNs) compared to ID-based recommendation models. The illustration, without considering node features, fails to adequately demonstrate the full capability of GNN models, particularly in highlighting their ability to leverage rich feature information.
2. Recently, some efforts not mentioned in this submission have been made to benchmark and evaluate existing work in continuous time domains, such as in [1]. Although this paper has a different focus, the range of GNN methods that can be evaluated is more extensive.

[1] Chen C, Geng H, Yang N, et al. Easydgl: Encode, train and interpret for continuous-time dynamic graph learning[J]. IEEE Transactions on Pattern Analysis and Machine Intelligence, 2024.

**Questions:**

See weakness for more details.

---

> ### Author Response · Authors · 2024-11-22
> **Response to Reviewer UMN2 (Part 1/2)**
>
> Thank you for these helpful comments. Our detailed answers are provided below.
> > W1: "*the node features are omitted.*"
>
> We agree that incorporating node features is one of the key advantages of GNNs compared to ID-based recommendation models. However, we would like to clarify that excluding features in the context of link prediction tasks is both a practical necessity and consistent with existing temporal graph benchmarks. In real-world dynamic graphs, features are often incomplete, noisy, and difficult to align across different types of nodes, especially in bipartite and heterogeneous graphs. As a result, link prediction models are often required to rely solely on interaction data (i.e., temporal edges with features). It is worth noting that existing temporal graph benchmarks typically lack features as well, primarily due to these practical challenges.
>
> Furthermore, interaction data is often more critical than feature data in link prediction tasks, as evidenced by prior research. For example, SGNN-HN is able to achieve excellent performance on the toy example in Figure 3 with an AP of 100% without any features. In contrast, existing temporal GNNs fail to capture the simple sequential dynamics in the toy example using only interaction data. This limitation warrants the community's attention. If temporal GNNs cannot effectively predict future links purely from interaction data, which is a common scenario in recommendation and many other real-world applications, they will face significant limitations in their applicability to practical tasks. This is also a key motivation behind our work.
>
> Last but not least, please note that existing temporal GNN studies [R2,R3] investigating the limitations of prior methods also do not take features into consideration, instead focusing on the modeling of temporal interaction data. This further validates our choice to exclude features in our toy example and highlights the importance of modeling interaction data in temporal graph learning.
>
> We provide detailed evidences and discussions below to support the above points.
>
> 1. **The commonly used datasets in temporal graph learning typically lack features.** All commonly used datasets in temporal graph learning (as summarized in Table 5) lack *node* features. As for *edge* features, among these 15 unique datasets, only Wikipedia and Reddit include 172-dimensional edge features. Other datasets have significantly lower-dimensional edge features, such as MOOC (4 dimensions), Social Evo. and tgbl-comment (2 dimensions), or even just a single dimension as in Flights, Contact, tgbl-review, and tgbl-coin.
> The feature data in real-world graphs is often incomplete, noisy, and difficult to align across different types of nodes. For example, the original GoogleLocal dataset contains the `price` feature for places, yet 267,200 out of 267,336 places lack this feature. Furthermore, the user features are personal information, while the place features are attributes of the venue. Aligning these features and learning a unified transformation for such different semantics is highly challenging for models. This difficulty in handling feature data also explains why datasets in the recommender systems literature often lack features and instead focus solely on user-item interaction data.
> 1. **The interaction data is more crucial than feature data in link prediction tasks.** Existing studies have shown that interaction data is more informative than feature data for link prediction tasks [R1]. To further validate this observation, we conducted experiments on GoogleLocal using semantic features, and the MRR (\%) results are as follows. For comparison, the original results (without features) are included from Table 3. Among the tested models, only JODIE, DyRep, and TGAT (with edge features only) show notable improvements when semantic features are included. Importantly, SGNN-HN, which does not use any features, still achieves the best performance among all temporal GNN models, including those using features. This highlights the critical importance of temporal interaction data for link prediction tasks and underscores the significant progress still needed for temporal graph learning models to fully leverage such data.
>
> ||JODIE|DyRep|TGAT|TGN|CAWN|TCL|GraphMixer|DyGFormer|
> |-|-|-|-|-|-|-|-|-|
> |original|36.84|28.77|19.49|**51.59**|**18.96**|**18.90**|20.32|**18.89**|
> |+ edge feat|**42.44**|**37.46**|17.35|48.20|15.76|8.58|21.38|18.53|
> |+ node feat|42.28|33.53|**30.50**|47.54|15.66|14.46|**21.51**|17.91|
>
> [R1] Weilin Cong, Si Zhang, Jian Kang, Baichuan Yuan, Hao Wu, Xin Zhou, Hanghang Tong, and Mehrdad Mahdavi. Do we really need complicated model architectures for temporal networks?
>
> [R2] Yanbang Wang, Yen-Yu Chang, Yunyu Liu, Jure Leskovec, and Pan Li. Inductive representation learning in temporal networks via causal anonymous walks.
>
> [R3] Luo Y, Li P. Neighborhood-aware scalable temporal network representation learning.

---

> ### Author Response · Authors · 2024-11-22
> **Response to Reviewer UMN2 (Part 2/2)**
>
> > W2. "*Recently, some efforts not mentioned in this submission have been made to benchmark and evaluate existing work in continuous time domains, such as in [1]. Although this paper has a different focus, the range of GNN methods that can be evaluated is more extensive.*"
>
> We appreciate the reviewer's acknowledgement of relevant works. We will include [1] in our revision. To extend the scope of our evaluation, we try to incorporate methods from other categories as [1] does, such as the static GNN model GCN and the general recommendation model LightGCN. The results of GCN and LightGCN on GoogleLocal are as follows:
> ||GoogleLocal|
> |-|-|
> |GCN|17.83|
> |LightGCN|32.16|
> |TGN|51.59|
> |SGNN-HN|**64.59**|
>
> These results indicate that static GNN models and general recommendation models (which do not account for temporal information) are less effective at capturing the sequential dynamics of temporal graphs compared to temporal GNN models and sequential recommendation models.
>
> Please note that the primary focus of this paper is on the limitations of existing temporal GNNs and benchmark datasets. This is why we selected nine state-of-the-art temporal GNN models for evaluation (with SGNN-HN included solely for comparison purposes). If the reviewer has specific suggestions for additional models to include, we would be happy to consider them.
>
> ---
> **The feature processing of GoogleLocal**
>
> We provide a detailed description of the feature processing for GoogleLocal in our response to W2 below for your reference. Specifically, the original GoogleLocal dataset contains node features for users and places, such as `name`, `jobs`, `currentPlace`, `previousPlace`, `education` for users, and `name`, `price`, `address`, `hour`, `closed` for places. We treat `price` and `closed` as one-dimensional features, while the combination of other features is processed using SBERT to generate semantic embeddings. These semantic embeddings are then reduced in dimensionality via PCA, resulting in a final embedding dimension of 172. Similarly, the edge features are processed as follows: the dataset includes user reviews of places, which consist of `rating`, `review_text`, and `category`. The `rating` is treated as a one-dimensional feature, while `review_text` and `category` are combined and processed using SBERT. SBERT is chosen for its ability to capture the semantic information of multilingual text, which is essential as the text in GoogleLocal is multilingual.

---

> > ### Comment · Reviewer_UMN2 · 2024-11-22
> >
> > Thanks for the authors' response and appreciate for the extensive experiments in the paper and rebuttal. I would raise my score to 6.

---

> > > ### Author Response · Authors · 2024-11-24
> > >
> > > We sincerely thank you for your positive evaluation and for recognizing the value of our work. Your insightful comments have been instrumental in improving our manuscript. We truly appreciate the time and effort you dedicated to reviewing our submission.

---

### Official Review · Reviewer_5t8Y · 2024-11-01

**Soundness:** 3
**Presentation:** 3
**Contribution:** 3
**Rating:** 8
**Confidence:** 5

**Summary:**

In this work, the authors pointed out that existing methods are lacking in the learning of sequential dynamics while focusing primarily on predicting repeated edges. To this end, the authors first demonstrated that methods such as GraphMixer and DyGformer are unable to learn sequential dynamics in a toy dataset and then introduce a new benchmark, TGB-Seq, that contains a minimal amount of repeated edges. TGB-Seq contains large real-world datasets that spans diverse domains as well as both bipartite and non-bipartite networks.

**Strengths:**

This paper focuses on examining how well current models can learn sequential dynamics (predict unseen edges). I believe the limitations of existing methods pointed out in this work is quite interesting. Here are the strengths of the paper:

- the idea of a benchmark containing datasets with very low amounts of repeated edges are interesting while opening up new research directions that design novel methods for such datasets. This is significant for the temporal graph learning community.

- The paper is well-written and clearly presented.

- The new datasets spans a diverse range of domains and contains both bipartite and non-bipartite networks. The authors also provided code and dataset access.

**Weaknesses:**

Here are some weakness of the paper:

- **new but familiar idea**. The idea of measuring unseen edges and their effect in model performance is not entirely new. In the TGB work, the authors also mentioned the surprise index which measures the amount of test edges unseen during training (similar to the unseen edges idea in the paper though different).  The authors should further clarify any definition difference between the "unseen" edges as shown In Figure 2 versus the novel edges $E_{test} \ E_{train}$ as mentioned in the surprise index.

- **lack of node, edge features for the datasets**. The authors mentioned that this work excludes datasets with node and edge features. Were there datasets that you considered that has node or edge features? Is it possible to include some features for the datasets in TGB-seq benchmark? Because node / edge features might be crucial for the model to learn sequential dynamics. For example, one user might enjoy coffee and this user will choose to purchase a new type of coffee beans (a categorical feature of the item node)) at a future time. With node features on the items, the model can learn this dynamics easier while it might be difficult to learn if no feature is provided.


- **OOT errors are not intuitive**. In terms of running time, memory based models such as TGN and DyRep are usually more efficient than DyGFormer and CAWN (such as compared in the TGB work). It is not very intuitive to me why these models can not finish a single epoch in 24 hours as they are able to scale to the largest datasets on TGB. More clarification / explanation from the authors are appreciated.

Considering that this work presents challenging new datasets, I am giving a positive score. However, I would like to know the answer to the concerns above.

**Questions:**

See weakness for the questions as well.

**Details Of Ethics Concerns:**

no ethical concerns.

---

> ### Author Response · Authors · 2024-11-22
> **Response to Reviewer 5t8Y (Part 1/3)**
>
> Thank you for these helpful comments. Our detailed answers are provided below.
> > W1. "*The idea of measuring unseen edges and their effect in model performance is not entirely new ... The authors should further clarify any definition difference between the "unseen" edges as shown In Figure 2 versus the novel edges as mentioned in the surprise index*"
>
> Thank you for the suggestion. The surprise index is defined as $\frac{|E_{test} \backslash E_{train}|}{|E_{test}|}$, which represents the ratio of new edges in the test set that do not appear in the training set. In our paper, unseen edges are defined as $\mathcal{E} \backslash \mathcal{E}\_{\rm seen}$, where an edge $e\_i=(s\_i,d\_i,t\_i)\in\mathcal{E}\_{\rm seen}$ if there exists an edge $e_j=(s_j,d_j,t_j)$ such that $s_i=s_j,d_i=d_j,t_j<t_i$. In other words, all edges appearing for the first time in the dataset are considered unseen edges. Kindly note that we have provided definition of the repeat ratio at Line 315 in our manuscript, i.e., $\frac{\mathcal{E}_{\rm seen}}{\mathcal{E}}$. We do not directly adopt surprise index in TGB and introduce unseen edges and the repeat ratio because the concept of surprise index does not fully align with our motivation. To elaborate, let us consider an illustrative extreme case:
>
> Suppose the training set only contains repeated instances of edge (a,b), and the test set only contains repeated instances of edge (b,c). The surprise index would be 1 since (b,c) is absent in the training set, while the repeat ratio would be very large, as the dataset consists of only two unique edges. Despite the high surprise index, existing temporal GNN models are likely to achieve good performance in this scenario. This is because they can leverage all historical edges (including those in the test set) prior to each prediction timestamp. The only challenging prediction would be the first instance of (b,c). Once (b,c) has been observed in the test set, subsequent predictions become significantly easier. This scenario highlights a limitation of the surprise index: it does not account for the temporal availability of historical edges in temporal GNN models, making it less effective at reflecting the challenge of predicting truly new edges in datasets.
>
> However, our focus in this paper is exactly on the challenge of predicting truly new edges that have never appeared before. Therefore, we introduce unseen edges and repeat ratio to precisely reflect the level of difficulty posed by such predictions.
> Though TGB measures datasets with surprise index and notes that a high surprise index implies greater difficulty based on experimental findings, its primary focus is not on proposing datasets with a high surprise index. Instead, as emphasized in the remark at Line 318 of our manuscript, TGB is designed to emphasize large graphs, diverse domains, and multiple negative evaluations to challenge existing methods. In fact, only two datasets in TGB exhibit a high surprise index (also low repeat ratio, as shown in Table 5).
> In our work, we propose datasets with low repeat ratio to address the gap that existing benchmarks exhibit excessive repeated edges, which cannot adequately assess models' ability to capture complex sequential dynamics. We believe that TGB-Seq serves as a crucial complement to existing benchmarks, enabling a more comprehensive evaluation of temporal GNN models.

---

> ### Author Response · Authors · 2024-11-22
> **Response to Reviewer 5t8Y (Part 2/3)**
>
> > W2. "*lack of node, edge features for the datasets*"
>
> The original datasets from recommender systems, such as GoogleLocal, ML-20M, Taobao, and Yelp, include additional text features. However, the non-bipartite datasets, such as YouTube, Patent, WikiLink, and Flickr, do not. We acknowledge that features might help improve the performance of models, and we have augmented GoogleLocal with semantic features for this purpose. Specifically, the original GoogleLocal dataset contains node features for users and places, such as `name`, `jobs`, `currentPlace`, `previousPlace`, `education` for users, and `name`, `price`, `address`, `hour`, `closed` for places. We treat `price` and `closed` as one-dimensional features, while the combination of other features is processed using SBERT to generate semantic embeddings. These semantic embeddings are then reduced in dimensionality via PCA, resulting in a final embedding dimension of 172. Similarly, the edge features are processed as follows: the dataset includes user reviews of places, which consist of `rating`, `review_text`, and `category`. The `rating` is treated as a one-dimensional feature, while `review_text` and `category` are combined and processed using SBERT. SBERT is chosen for its ability to capture the semantic information of multilingual text, which is essential as the text in GoogleLocal is multilingual.
>
> The MRR (%) results on the augmented GoogleLocal dataset are as follows. The original results are copied from Table 3 for comparison. Notably, only the results for JODIE, DyRep, and TGAT (with edge features only) show significant improvement. This might be due to the fact that the node features of users and places are not well aligned. For instance, the last two dimensions of the place features correspond to `rating` and `closed`, while the user features consist entirely of semantic embeddings derived from personal information. This misalignment may make it challenging for the models to learn a unified transformation for these features.
>
> ||JODIE|DyRep|TGAT|TGN|CAWN|TCL|GraphMixer|DyGFormer|
> |-|-|-|-|-|-|-|-|-|
> |original|36.84|28.77|19.49|**51.59**|**18.96**|**18.90**|20.32|**18.89**|
> |+ edge feat|**42.44**|**37.46**|17.35|48.20|15.76|8.58|21.38|18.53|
> |+ node feat|42.28|33.53|**30.50**|47.54|15.66|14.46|**21.51**|17.91|
>
> Though feature data may help in learning sequential dynamics, we would like to clarify that link prediction models are often required to rely solely on interaction data in real-world scenarios. On one hand, **features in real-world dynamic graphs are often incomplete, noisy, and difficult to align across different types of nodes, particularly in bipartite and heterogeneous graphs**. For example, in GoogleLocal, user features and place features cannot be aligned due to their entirely different semantic meanings, as discussed earlier. Additionally, we observe that 267,200 out of 267,336 places in GoogleLocal lack the price feature entirely. These practical challenges also lead to the absence of features or the use of only low-dimensional features in many existing temporal graph benchmarks. On the other hand, **interaction data is often more crucial than feature data in link prediction tasks.** Notably, SGNN-HN, which does not use any features, still achieves the best performance among all temporal GNN models on GoogleLocal, including those using features. This highlights the critical importance of temporal interaction data for link prediction tasks and underscores the significant progress still needed for temporal graph learning models to fully leverage such data. Therefore, we believe that the TGB-Seq datasets effectively reflect real-world challenges in temporal graph learning and open up new research directions, even in the absence of features.
>
> That said, we will process the features of ML-20M, Taobao and Yelp in a similar manner as well, and make the processed features and the cleaned raw text features public. And we will report the results with node/edge processed feature in our revision. If you have further suggestions or ideas, we would greatly appreciate your feedback and welcome further discussion.

---

> ### Author Response · Authors · 2024-11-22
> **Response to Reviewer 5t8Y (Part 3/3)**
>
> > W3. "*OOT errors are not intuitive*"
>
> We sincerely thank you for identifying this issue. After examining the differences between the TGN/DyRep implementations in our adopted library, DyGLib, and those in TGB, we found that DyGLib incurs significantly higher training time than TGB for memory-based methods, including JODIE, TGN, and DyRep.
>
> To clarify, we first review the training process of memory-based methods for a batch sample, which consists of the following steps:
> 1. compute the updated memory for nodes using raw messages
> 2. Aggergating the information from neighbors (including their memory) to compute node embeddings for positive and negative samples at current batch
> 3. Update the memory of the positive samples, and then replace their previous raw messages with the new messages received from the current batch
>
> The primary reasons for the significant time difference between DyGLib and TGB are as follows:
> 1. **Memory updates for all nodes vs. selective nodes.** At Step 1, DyGLib computes the updated memory for all nodes with raw messages, whereas TGB computes the memory only for nodes involved in the current batch, including the source and destination nodes in the positive and negative samples, as well as their $k$-hop neighbors. Here, $k$ is commonly set to 1.
>     Computing the updated memory for all nodes is time-intensive, as the number of nodes with raw messages progressively increases during training. This occurs because a node retains its raw messages from the last batch in which it was a positive sample. Consequently, once a node has its first interaction, it retains raw messages until the end of the training epoch. Therefore, DyGLib is required to update the memory for nearly all nodes in the dataset in the later stages of an epoch, significantly increasing the time cost.
>
> 2. **The data structure for storing raw messages matters.** DyGLib allows storing multiple raw messages for each node, while TGB only retains the last raw message. Selecting the last raw message to update memory is a common practice in memory-based methods and typically achieves better performance than aggregating multiple messages (e.g., using their mean). Therefore, TGB's simpler implementation is sufficient for practical use. This distinction leads to a significant time gap between DyGLib and TGB. Specifically, DyGLib uses a Python dict to store raw messages, where the structure for each node is represented as a list, i.e., `{'node 1': [raw message 1, raw message 2, ...], 'node 2': [raw message 1, raw message 2, ...]}`. TGB also uses a Python dict for raw messages but fixes the raw message as a tuple and initializes the raw message for all nodes to zero at the start of each epoch. When a node's raw message is updated, TGB simply reassigns its value, keeping the size of the dict constant throughout the epoch. In contrast, DyGLib frequently removes and appends raw messages during Step 3. This operation is time-consuming due to the inefficiencies inherent in Python's dict and list implementations. As a result, DyGLib processes raw messages with significantly higher time costs compared to TGB.
>
> Note that these above issues become severe only when training on large datasets. For small datasets, memory-based methods implemented by DyGLib are more efficient than CAWN and TGB.
>
> Based on the above observations, we revise DyGLib as follows: we use a tensor to store raw messages and only retain the last raw message for each node, and we compute the updated memory only for nodes involved in the current batch. These revisions significantly reduce the time cost of memory-based methods in DyGLib. As of this response, we have completed experiments on GoogleLocal, ML-20M, and YouTube. Notably, we re-ran the GoogleLocal experiments because the training time exceeded two days in the previous experiments. The updated MRR (%) results are as follows.
>
> ||GoogleLocal|ML-20M|YouTube|
> |-|-|-|-|
> |JODIE|42.24|20.45|64.83|
> |DyRep|36.69|21.29|65.19|
> |TGN|54.43|21.48|72.68|
>
> The average training time per epoch for JODIE, DyRep, and TGN on GoogleLocal is now 149s,198s,336s, respectively. Experiments on other datasets are ongoing, and we will update Figure 5, Table 3, and Table 4 in our revision accordingly. We will release the updated version of our code for public use and notify the DyGLib developers about this issue, with the hope of benefiting the community.

---

> > ### Comment · Reviewer_5t8Y · 2024-11-22
> > **Reviewer Reply to Author Response**
> >
> > Thanks for addressing my concerns and I believe this benchmark will be beneficial to the community. Therefore, I will increase my score to 8.

---

> > > ### Author Response · Authors · 2024-11-24
> > >
> > > Thank you for your positive feedback. We greatly appreciate your recognition of our contributions and the constructive suggestions that have helped us refine our manuscript. Once again, thank you for your time and effort in reviewing our submission.

---

### Official Review · Reviewer_GU3D · 2024-11-04

**Soundness:** 3
**Presentation:** 3
**Contribution:** 3
**Rating:** 6
**Confidence:** 2

**Summary:**

TGB-Seq presents a novel benchmark for the temporal link prediction task by emphasizing that the key characteristic of temporal link prediction is that the model should learn how to rank the most likely destination node for a given source node at the queried time, which aligns well with practical, real-world settings, such as recommender systems.

**Strengths:**

1. Incorporates diverse datasets from different domains and different sizes (number of nodes/edges).
2. Comprehensive report for experimental settings (hyperparameters search range, computational resources, etc.)  and assessment of SOTA Temporal GNNs baselines.
3. Clear presentation, and the paper is easy to read.

**Weaknesses:**

1. For validation and test set, TGB-Seq only retains nodes that appear in the training set. I do not think this setting is practical for real-world temporal graphs, as they continuously evolve, and this setting is not comprehensive enough to assess the model’s ability to reason on new information that emerges in future timestamps.

2. Previous work [1] (Poursafaei et al., 2022) also devised 2 other negative sampling strategies (NSS): (1) random NSS, which samples any possible node pairs, and (2) inductive NSS that samples edges that are unseen during the training stage. Therefore, previous works' inductive NSS also do not rely on historical edges, and random NSS samples are also from all possible nodes. How does TGB-Seq’s NSS differ from [1]’s inductive or random NSS?

3. TGB-Seq claims that the proposed evaluations of SOTA models show that achieving both efficiency and effectiveness in temporal GNNs remains an open problem, highlighting the distinctive feature of TGB-Seq. However, TGB [2] also presents large temporal graphs that could challenge any model’s efficiency, and some of their datasets have high surprise scores, which also challenge the performance of SOTA models that are listed in TGB-Seq. Compared to TGB [2], how distinctive is TGB-Seq’s capabilities in challenging Temporal GNNs?

Reference:

[1] Towards better evaluation for dynamic link prediction.

[2] Temporal Graph Benchmark for Machine Learning on Temporal Graphs

**Questions:**

Please refer to Weaknesses.

---

> ### Author Response · Authors · 2024-11-22
> **Response to Reviewer GU3D (Part 1/2)**
>
> Thank you for these helpful comments. Our detailed answers are provided below.
>
> > W1. "*For validation and test set, TGB-Seq only retains nodes that appear in the training set. I do not think this setting is practical for real-world temporal graphs, as they continuously evolve, and this setting is not comprehensive enough to assess the model’s ability to reason on new information that emerges in future timestamps.*"
>
> We appreciate your concern and agree that real-world graphs are dynamic and continuously evolving. However, our chosen setting aligns with the primary focus of this paper. Specifically, predicting interactions for new nodes, where no historical data is available, represents a cold-start problem that is inherently more challenging than predicting interactions for existing nodes with observed interactions.
>
> In the recommendation literature, the cold-start problem is widely discussed and often treated as a separate research problem [R1,R2]. In general recommendation studies, the cold-start problem is typically excluded to focus on the core task of recommending items to users based on their historical preferences—such as in sequential recommendation and session-based recommendation. They exclude users and items with fewer than a certain number of interactions and conduct evaluations only on users and items with observed interactions [R3].
>
> Similarly, in this work, we focus on the core problem of predicting interactions for existing nodes with observed interactions. As demonstrated in our motivations and toy example, this problem is challenging and requires substantial effort to achieve significant success.
> That said, we acknowledge that predicting interactions for new nodes is an important and valuable research direction, and we plan to consider it in our future work.
>
> [R1] Schein A I, Popescul A, Ungar L H, et al. Methods and metrics for cold-start recommendations.
>
> [R2] Lam X N, Vu T, Le T D, et al. Addressing cold-start problem in recommendation systems.
>
> [R3] Kang W C, McAuley J. Self-attentive sequential recommendation.
>
> > W2. "*Previous work [1] (Poursafaei et al., 2022) also devised 2 other negative sampling strategies (NSS): (1) random NSS, which samples any possible node pairs, and (2) inductive NSS that samples edges that are unseen during the training stage. Therefore, previous works' inductive NSS also do not rely on historical edges, and random NSS samples are also from all possible nodes. How does TGB-Seq's NSS differ from [1]'s inductive or random NSS?*"
>
> The NSS used in TGB-Seq is equivalent to random NSS with collision checks. This approach aligns with the goals of the TGB-Seq dataset. Let $E_{\rm all}, E_{\rm train}, E_{\rm teset}$ denote the set of all edges in the dataset, the training set and the test set, respectively. Let $U$ denote the set of all possible node pairs, and $E_t$ the set of arriving edges at time $t$. Our NSS is randomly sampled from $U \bigcap \overline{E_t}$, the same as that of random NSS in [1].
> [1] proposed random NSS, historical NSS, and inductive NSS. The historical NSS samples negative edges from $E_{\rm train} \bigcap \overline{E_t}$ to evaluate whether a given method is able to predict in which timestamps an seen edge would reoccur. Inductive NSS, while similar to the idea of historical NSS, focuses on the test set. It samples negative edges from $E_{\rm test} \bigcap \overline{E_{\rm train}} \bigcap \overline{E_t}$ to evaluate the reocurrence pattern of edges only seen during test time.
> These two settings are not suitable for TGB-Seq datasets, as they focus on challenging the model with reoccurrence pattern of seen edges in the training set or the test set. However, TGB-Seq datasets are designed to emphasize unseen edges, which have a low ratio of seen edges. Using historical NSS or inductive NSS would not increase the difficulty of the task, but would introduce bias to the evaluation. Therefore, we adopt random NSS with collision check to align with TGB-Seq's motivation.
>
> Additionally, please note that we use multiple negative samples for each positive edge. This approach provides a sufficiently challenging evaluation, addressing the same concern that historical NSS and inductive NSS were designed to handle, that is, the oversimplistic evaluation arising when there is only a single negative sample. Our approach is consistent with the existing TGB benchmark [2].

---

> ### Author Response · Authors · 2024-11-22
> **Response to Reviewer GU3D (Part 2/2)**
>
> > W3. "*TGB-Seq claims that the proposed evaluations of SOTA models show that achieving both efficiency and effectiveness in temporal GNNs remains an open problem, highlighting the distinctive feature of TGB-Seq. However, TGB [2] also presents large temporal graphs that could challenge any model’s efficiency, and some of their datasets have high surprise scores, which also challenge the performance of SOTA models that are listed in TGB-Seq. Compared to TGB [2], how distinctive is TGB-Seq’s capabilities in challenging Temporal GNNs?*"
>
> TGB-Seq is carefully curated to provide a comprehensive benchmark for **evaluating temporal GNNs on real-world datasets that inherently exhibit complex sequential dynamics**. To achieve this, we include datasets from recommender systems, social networks, citation networks, and web link networks. These real-world scenarios demand that models effectively capture underlying sequential dynamics to make accurate predictions.
>
> While TGB includes two datasets with high surprise scores (tgbl-review and tgbl-comment), the remaining datasets in TGB exhibit relatively fewer unseen edges (i.e., high repeat ratios), as shown in Table 5 of our manuscript. Consequently, these two datasets alone are insufficient to comprehensively evaluate the ability of temporal GNNs to capture sequential dynamics, both in terms of the number of datasets and domain diversity.
> In other words, TGB does not address the gap caused by excessive edge repetition in existing benchmark datasets, which limits their ability to assess models' capacity to learn sequential dynamics. In fact, TGB's primary focus is not on proposing datasets with a high surprise index. Instead, as noted in the remark at Line 318 of our manuscript, TGB emphasizes large graphs, diverse domains, and multiple negative evaluations to challenge existing methods. In contrast, TGB-Seq addresses this gap by providing datasets explicitly designed to evaluate temporal GNNs on their ability to handle sequential dynamics across diverse, real-world scenarios.

---

> > ### Comment · Reviewer_GU3D · 2024-11-25
> >
> > Thanks for the answer. After reading it, I will keep my positive rating.

---

### Official Review · Reviewer_wMVB · 2024-11-05

**Soundness:** 3
**Presentation:** 4
**Contribution:** 3
**Rating:** 6
**Confidence:** 4

**Summary:**

This paper introduces TGB-Seq, a new benchmark designed to challenge temporal GNNs with complex sequential dynamics. Existing datasets often have excessive repeated edges, which fail to capture the sequential patterns present in many real-world applications like recommender systems and social networks. TGB-Seq minimizes repeated edges and includes diverse domains such as e-commerce, movie ratings, and social networks. The study reveals that current temporal GNN methods struggle with these datasets, highlighting the need for models that can better generalize to unseen edges. The paper contributes by providing new datasets that emphasize sequential dynamics and exposing the limitations of existing GNN approaches.

**Strengths:**

1. This article highlights the issue of excessive repeated edges in existing dynamic graph datasets. The experiments presented in the text are very complete and convincing, and the presentation in Figure 2 is impressive. The motivation behind the construction of TGB-Seq is very sufficient, the problems it addresses are very clear, and it has significant practical implications.
2. The structure of this article is clear, and the expression is explicit. Overall, it is easy to follow.

**Weaknesses:**

1. Section 3.2 is far from satisfactory. The conclusion of Section 3.2 is very interesting and quite important, but the entire section lacks formal expression, making the analysis of the limitations of existing dynamic graph models in this part very perfunctory.
2. The assumptions and premises in Section 3.2 are not clear. There is a simple description of premises starting from line 212 in the text, but it is very vague. Are the node features of the three types of nodes u, v, and i all blank? If so, this assumption lacks rationality. The subsequent discussions on the model's memory and aggregation parts are all based on this. If all nodes have the same features (all nodes have no features), it will inevitably lead to the indistinguishability of many nodes, which is similar to the discussion of graph isomorphism problems. In such an extreme case, the limitations of dynamic graph models are difficult to be widely recognized.
3. Continuing from the previous point, the authors should discuss why SGNN-HN can achieve such good results under the toy example and what problems it overcomes. Of course, these contents should be formally expressed in the form of expressions. The existing theoretical discussion in the text is difficult to understand.

**Questions:**

Please see my comments for details

---

> ### Author Response · Authors · 2024-11-22
> **Response to Reviewer wMVB (Part 1/3)**
>
> Thank you for these helpful comments. Our detailed answers are provided below.
> > W1. "*Section 3.2 is far from satisfactory. The conclusion of Section 3.2 is very interesting and quite important, but the entire section lacks formal expression, making the analysis of the limitations of existing dynamic graph models in this part very perfunctory.*"
>
> We appreciate your suggestion and have revised Section 3.2 in our updated PDF as follows:
> 1. Added formal expressions to describe the memory modules and aggregation modules.
> 2. Discussed the typical implementations of these two modules in existing temporal graph models.
> 3. Re-wrote the analysis with more detailed explanations and examples to better illustrate the limitations of existing models.
>
> Please review the revised Section 3.2, with changes highlighted in blue for your convenience. If you have any further suggestions, please let us know.

---

> ### Author Response · Authors · 2024-11-22
> **Response to Reviewer wMVB (Part 2/3)**
>
> > W2. "*Are the node features of the three types of nodes u, v, and i all blank? If so, this assumption lacks rationality.*"
>
> Yes, the node features of nodes u, v, and i are all blank in the toy example. However, this assumption in the context of link prediction tasks is both a practical necessity and consistent with existing temporal graph benchmarks.
> **First**, the toy example without features still effectively reveals the limitations of existing dynamic graph models. Notably, **SGNN-HN, which also does not use any features, achieves excellent performance on the toy example with an AP of 100%**. This demonstrates that it is possible to learn the underlying sequential dynamics in the toy example without relying on features, yet existing temporal GNN models fail to do so. This underscores the limitations of existing models in capturing sequential dynamics purely from interaction data (i.e., temporal edges with features).
> **Second**, it is important to note that modeling sequential dynamics solely from interaction data is critical for practical link prediction tasks. In real-world dynamic graphs, features are often incomplete, noisy, and difficult to align across different types of nodes, especially in bipartite and heterogeneous graphs. As a result, link prediction models are often required to rely solely on interaction data. Existing temporal graph benchmarks typically lack features as well, primarily due to these practical challenges.
> **Third**, interaction data is often more crucial than feature data in link prediction tasks, as evidenced by prior research.
> **Fourth**, existing temporal GNN studies [R2,R3] investigating the limitations of prior methods also do not take features into consideration, focusing on the modeling of temporal interaction data.
>
> Therefore, we exclude features in the toy example to create a clean, illustrative case to highlight the limitations of existing temporal GNNs. We elaborate on the second and third points with detailed examples below.
>
> 1. **Real-world dynamic graph datasets typically lack features.** The commonly used datasets in temporal graph learning (as summarized in Table 5) generally lack *node* features. As for *edge* features, among these 15 unique datasets, only Wikipedia and Reddit include 172-dimensional edge features. Other datasets have significantly lower-dimensional edge features, such as MOOC (4 dimensions), Social Evo. and tgbl-comment (2 dimensions), or even just a single dimension as in Flights, Contact, tgbl-review, and tgbl-coin.
> This lack of feature data is a common characteristic of real-world dynamic systems because feature data is often incomplete, noisy, and difficult to align across different types of nodes, especially in bipartite and heterogeneous graphs. For example, the original GoogleLocal dataset contains the `price` feature for places, yet 267,200 out of 267,336 places lack this feature. Furthermore, the user features are personal information, while the place features are attributes of the venue. Aligning these features and learning a unified transformation for such different semantics is highly challenging for models. This difficulty in handling feature data also explains why datasets in the recommender systems literature often lack features.
> 1. **The interaction data is more crucial than feature data in link prediction tasks.** Existing studies have shown that interaction data is more informative than feature data for link prediction tasks [R1]. To further validate this observation, we conducted experiments on GoogleLocal using semantic features, and the MRR (%) results are as follows. For comparison, the original results (without features) are included from Table 3. Among the tested models, only JODIE, DyRep, and TGAT (with edge features only) show notable improvements when semantic features are included. Importantly, SGNN-HN, which does not use any features, still achieves the best performance among all temporal GNN models, including those using features. This highlights the critical importance of interaction data for link prediction tasks and underscores the significant progress still needed for temporal graph learning models to fully leverage such data.
>
> ||JODIE|DyRep|TGAT|TGN|CAWN|TCL|GraphMixer|DyGFormer|
> |-|-|-|-|-|-|-|-|-|
> |original|36.84|28.77|19.49|**51.59**|**18.96**|**18.90**|20.32|**18.89**|
> |+ edge feat|**42.44**|**37.46**|17.35|48.20|15.76|8.58|21.38|18.53|
> |+ node feat|42.28|33.53|**30.50**|47.54|15.66|14.46|**21.51**|17.91|
>
> [R1] Weilin Cong, Si Zhang, Jian Kang, Baichuan Yuan, Hao Wu, Xin Zhou, Hanghang Tong, and Mehrdad Mahdavi. Do we really need complicated model architectures for temporal networks?
>
> [R2] Yanbang Wang, Yen-Yu Chang, Yunyu Liu, Jure Leskovec, and Pan Li. Inductive representation learning in temporal networks via causal anonymous walks.
>
> [R3] Luo Y, Li P. Neighborhood-aware scalable temporal network representation learning.

---

> ### Author Response · Authors · 2024-11-22
> **Response to Reviewer wMVB (Part 3/3)**
>
> > W3. "*Continuing from the previous point, the authors should discuss why SGNN-HN can achieve such good results under the toy example and what problems it overcomes. Of course, these contents should be formally expressed in the form of expressions. The existing theoretical discussion in the text is difficult to understand.*"
>
> Thank you for this interesting question. We are currently investigating this topic, and our recent findings suggest the following potential reasons:
> 1. SGNN-HN maintains a learnable embedding for each node, which provides a strong representation capacity.
> 2. SGNN-HN effectively leverages historical neighbors to model sequential dynamics, allowing it to learn meaningful and distinctive embeddings for each node.
>
> Therefore, with these distinctive embeddings for nodes, SGNN-HN successfully distinguishes $i_4$ and $i_9$, which is key to addressing the challenge in the toy example. We elaborate on SGNN-HN's design and its advantages below.
>
> First, we observe that one of the main differences between SGNN-HN and existing temporal GNN models is that SGNN-HN maintains a learnable embedding ${\bf{x}}$ for each node, rather than relying on fixed node features. These embeddings are directly updated as model paramters during training. We believe this design allows SGNN-HN to learn more expressive representations for nodes, which is crucial for distinguishing nodes in the toy example.
>
> Moreover, SGNN-HN models sequential dynamics using historical neighbors in an intricate way. Suppose the historical neighbors of node $s$ are $d_1,\ldots, d_m$ in temporal order, and their embeddings are denoted as ${\bf{x}}\_1, {\bf{x}}\_2, \ldots, {\bf{x}}\_m$, respectively, where $m$ is the maximum number of historical neighbors considered. SGNN-HN constructs a virtual graph as follows. First, it introduce a virtual node, referred to as the *star node*. Its initial embedding is set as the mean of the neighbor embeddings: $\frac{1}{m}\sum\_{i=1}^m {\bf x}\_i$. Next, the following edges are added to the virtual graph: 1. sequential edges between neighbors, $(d_i,d_{i+1})$ for $1\le i<m$. 2. the bidirectional edges between the star node and each neighbor $d_i$ for $1\le i\le m$. SGNN-HN then applies an $\ell$-layered GNN to the virtual graph to obtain hidden representations for nodes, $\\{{\bf{h}}_i\\}$. (This GNN employs multiple advanced techniques, such as gating and attention mechanisms, to fully utilize the historical information. We do not expand on these details here because they are not important for understanding the main idea, but we are happy to provide more information if needed.) This design of the virtual graph, combined with the use of GNN, builds strong connections between historical neighbors while reducing the similarity between nodes that have never interacted. In the toy example, $i_4$ and $i_9$ have entirely distinct neighborhoods with no overlap. Consequently, SGNN-HN learns completely different embeddings for these two nodes, enabling it to effectively distinguish between them.
>
> It is worth noting that the ability of SGNN-HN to capture complex sequential dynamics, while existing temporal GNN models cannot, is key to addressing challenges in our TGB-Seq datasets. However, fully understanding this capability is beyond the scope of this paper, as it remains an ongoing area of investigation. For now, we plan to provide a brief discussion of these ideas in the main text and include a more formal introduction of SGNN-HN in the appendix of a future version. If you have further suggestions or ideas, we would greatly appreciate your feedback and welcome further discussion.
>
> ---
> **The feature processing of GoogleLocal**
>
> We provide a detailed description of the feature processing for GoogleLocal in our response to W2 below for your reference. Specifically, the original GoogleLocal dataset contains node features for users and places, such as `name`, `jobs`, `currentPlace`, `previousPlace`, `education` for users, and `name`, `price`, `address`, `hour`, `closed` for places. We treat `price` and `closed` as one-dimensional features, while the combination of other features is processed using SBERT to generate semantic embeddings. These semantic embeddings are then reduced in dimensionality via PCA, resulting in a final embedding dimension of 172. Similarly, the edge features are processed as follows: the dataset includes user reviews of places, which consist of `rating`, `review_text`, and `category`. The `rating` is treated as a one-dimensional feature, while `review_text` and `category` are combined and processed using SBERT. SBERT is chosen for its ability to capture the semantic information of multilingual text, which is essential as the text in GoogleLocal is multilingual.

---

> ### Author Response · Authors · 2024-11-25
> **Summary and Looking Forward to Further Discussions**
>
> Thank you again for your great efforts and valuable comments. As the author-reviewer discussion phase comes to a close, we look forward to any additional feedback you may have. Below, we summarize our previous responses for your convenience:
>
> 1. W1: Regarding the writing of Section 3.2, we have revised the description of the toy dataset, added formal expressions, and analyzed the limitations of existing temporal GNNs with their typical implementations. These changes are highlighted in blue in the revised manuscript.
> 2. W2: Regarding the lack of features in the toy example, we have explained that the absence of features does not affect our analysis of existing temporal GNNs. We have also highlighted the practical necessity of conducting link prediction without features and its consistency with existing benchmarks and limitation analyses of temporal GNNs.
> 3. W3: Regarding SGNN-HN's superior performance on the toy example, we have discussed the potential reasons in detail.
>
> If you have any further questions or require additional clarification, we welcome continued discussion and engagement.

---

### Official Review · Reviewer_Pkns · 2024-11-05

**Soundness:** 3
**Presentation:** 3
**Contribution:** 3
**Rating:** 6
**Confidence:** 3

**Summary:**

The authors present a new, challenging benchmark for temporal graph modeling, specifically addressing scenarios with fewer or no repeated edges. They provide an analysis of the limitations of the existing methods and conduct experiments using both current temporal graph modeling methods and sequential recommendation methods.

**Strengths:**

1. The motivation behind this work is compelling, as it is meaningful and reasonable to consider the repeat behavior in the temporal graph.

2. The proposed benchmark spans multiple domains with different data sizes and includes both bipartite and non-bipartite graphs, offering diverse resources for the research community to investigate issues around unseen edges.

3. The paper is well-structured and easy to follow.

**Weaknesses:**

1. While the paper focuses on unseen edges in temporal graphs, the selected baselines are primarily general-purpose models within the graph or recommendation domains. There are specific works that address similar challenges. For example:

- In the temporal graph domain, there are models with the ability to handle the inductive nodes:

   - TREND: TempoRal Event and Node Dynamics for Graph Representation Learning. WWW’22.

    - On the Feasibility of Simple Transformer for Dynamic Graph Modeling. WWW’24

- In the recommendation domain, Models tailored to repeat and exploration behaviors, including:

    - RepeatNet: A Repeat Aware Neural Recommendation Machine for Session-Based Recommendation. AAAI’19
    - Repetition and Exploration in Sequential Recommendation. SIGIR’23
    - Right Tool, Right Job: Recommendation for Repeat and Exploration Consumption in Food Delivery. RecSys’24
    - To Copy, or not to Copy; That is a Critical Issue of the Output Softmax Layer in Neural Sequential Recommenders. WSDM’24

    - (the explore-only model) "Masked and Swapped Sequence Modeling for Next Novel Basket Recommendation in Grocery Shopping”, RecSys’23

Thus the review would like to see what the performance would be like using the specifically designed models on the new benchmark.

2. The toy example offers a controlled setting to assess sequential pattern recognition with the existing methods. However, it may oversimplify the problem, limiting its ability to reflect real-world dynamics. I think the authors should analyze the performance limitations in realistic settings to provide a more comprehensive understanding of the critical limitations of the existing methods, offering insights about the possible improvement areas.

3. The analysis in Tables 3 and 4 provides limited insights, focusing primarily on overall performance decreases with the new benchmark. Deeper insights should be included to inspire future model design for this benchmark. For example, evaluating models separately on seen and unseen edges to highlight how well they handle these distinct cases.

4. The authors claim that they include a state-of-the-art recommendation model named SGNN-HN. However, it was published in 2020, which is not the state-of-the-art recommendation model.

**Questions:**

The authors only use one metric MRR to measure the performance, what about other ranking metrics (such as NDCG, Recall)? Do we need to design other metrics to measure the performance?

---

> ### Author Response · Authors · 2024-11-22
> **Response to Reviewer Pkns (Part 1/3)**
>
> Thank you for these helpful comments. Our detailed answers are provided below.
>
> > W1. "*While the paper focuses on unseen edges in temporal graphs, the selected baselines are primarily general-purpose models within the graph or recommendation domains. There are specific works that address similar challenges. For example: ... Thus the review would like to see what the performance would be like using the specifically designed models on the new benchmark.*"
>
> We appreciate the reviewer's acknowledgement of relevant works. We provide additional experimental results for RepeatNet[AAAI'19] and GRU4RecCPR[WSDM'24] (To Copy, or not to Copy) on GoogleLocal and ML-20m as follows. We also include the corresponding performance of GRU4Rec, and SGNN-HN from Table 3 for easy comparison.
>
> |Datasets|ML-20M|GoogleLocal|
> |-|-|-|
> |RepeatNet|25.23|OOT|
> |GRU4Rec|32.14|46.76|
> |GRU4RecCPR|32.12|46.82|
> |SGNN-HN|**34.80**|**64.59**|
>
> We have not provided the results for other methods for the following reasons:
> 1. TREND: TREND is designed for handling inductive nodes, which is not the primary focus of our paper. Instead, we focus on inductive edges and have demonstrated the limitations of existing graph learning methods in predicting inductive edges using nine popular temporal GNN methods. That said, we are currently adapting TREND for DyGLib to ensure a fair comparison, which requires additional effort. We will update the results once these experiments are completed.
> 2. SimpleDyG[WWW'24] (On the Feasibility of Simple Transformer for Dynamic Graph Modeling), ExpRec[RecSys'24] (Right Tool, Right Job) and BTBR[RecSys'23] (Masked and Swapped Sequence Modeling for Next Novel Basket Recommendation in Grocery Shopping): These methods aim to predict *a set of nodes* linked to a query node at a given time, whereas our task focuses on predicting a single node linked to a query node in a temporal graph.
> 3. "Repetition and Exploration in Sequential Recommendation": This paper does not propose a new model but rather examines repetition and exploration in sequential recommendation. One of its key findings is that sequential recommendation methods perform better at predicting repeated interactions than at predicting unseen interactions. These findings align with our motivation, but our work focuses on temporal graph learning, whereas the referenced paper focuses on sequential recommendation.
>
> It is important to note that while several works address or discuss the repetition and exploration problem, they are primarily situated in the domains of next basket recommendation (NBR), session-based recommendation, and sequential recommendation (SR). In contrast, our work focuses on temporal graph learning. Thus, we select popular temporal GNN methods and evaluate them on our proposed benchmark to highlight the limitations of these methods. SGNN-HN, a sequential recommendation model, is included for comparison purposes to demonstrate that achieving better performance is possible on our datasets, yet existing temporal graph learning models fail to do so.
>
> Moreover, existing NBR and SR methods are not directly applicable to the domain of temporal graph learning due to several key differences:
> 1. NBR focuses on predicting a set of items for users, while temporal graph learning focuses on predicting a single destination node for a query source node in a temporal graph.
> 2. Many recommendation methods are designed for bipartite graphs without features or interaction timestamps, which limits their ability to fully leverage the information available in existing temporal graph datasets. For example, as shown in Table 3, SGNN-HN fails to outperform existing temporal GNN methods on Wikipedia and Reddit.
>
> Additionally, our primary motivation is to highlight the limitations of existing temporal graph learning methods, particularly their inability to effectively explore new edges. This motivation aligns with existing works in NBR and SR, which discover the importance of repetition and exploration. Therefore, we believe that our work complements existing research by providing a new perspective on the challenges of dynamic graph learning.

---

> > ### Comment · Reviewer_Pkns · 2024-11-24
> >
> > Thank you to the authors for their efforts in providing detailed explanations and conducting the experiments.
> >
> >  I would like to discuss **single destination node VS a set of nodes**:  The authors claim they focus on predicting a single destination node. However, I have two concerns:
> >
> > (1) I think the related works that predict a set of nodes could potentially be adapted to the single-node setting. Therefore, the authors should reconsider the stated limitations of existing temporal graph methods (such as GNN-based and Transformer-based), particularly in handling unseen edges. Are these methods truly incapable of addressing such scenarios, or could they be extended to do so?
> >
> >  (2) I think predicting a set of nodes seems more reasonable in real-world scenarios for the temporal graph learning task.  In practice, we often don't know the exact timing of when a node will connect with others. Predicting a set of nodes, or the next K steps, offers greater flexibility and better aligns with real-world dynamics.

---

> > > ### Author Response · Authors · 2024-11-24
> > > **Further Response to Reviewer Pkns**
> > >
> > > Thank you for your valuable feedback.
> > >
> > > Regarding the concern (1): We agree that the related works could potentially be adapted to the single-node setting. For SimpleDyG [WWW'24], which predicts a sequence of potential destination nodes for a given source node, we will adapt it to the single-node setting by considering only the first destination node as the target node.
> > > Additionally, we have completed the adaptation of TREND [WWW'22] into DyGLib, and the experiments are currently running. **We will include the results of both SimpleDyG and TREND in our revised manuscript.** We believe these additions will make our paper a more comprehensive evaluation of existing methods. Thank you for informing us about these related works again!
> > >
> > > Furthermore, **we have revised our manuscript as follows**:
> > > 1. We introduce TREND and SimpleDyG when discussing existing temporal GNNs in Section 2.
> > > 2. We include a discussion on repeat and exploration behaviors in recommender systems in Section 2, incorporating the studies you mentioned.
> > >
> > > Please check the revised Section 2, "Related Work", in our updated manuscript PDF, with changes highlighted in red for your convenience.
> > >
> > > ---
> > >
> > > Regarding the concern (2): We agree that predicting a set of nodes is more reasonable in real-world scenarios. This also highlights that more significant progress is still needed for temporal graph learning models to better align with real-world tasks. This is precisely why we proposed the TGB-Seq benchmark, as a first step toward providing a comprehensive benchmark that reflects the challenges of real-world sequential dynamics.
> > >
> > > In this paper, we chose to evaluate models in the single-node setting because most existing temporal GNNs are designed for this setting. Evaluating these methods in the single-node setting allows us to fairly reveal their limitations. That said, our TGB-Seq datasets can easily be adapted to evaluate set-of-nodes prediction tasks. For instance, we could merge the destination nodes associated with each source node in the test set and treat them as the test ground truth for that node. The model would then be tasked with ranking these positive destination nodes along with sampled negative destination nodes. Model developers can select the appropriate evaluation methods based on their specific tasks, and we will consider incorporating this evaluation setting in our future work.
> > >
> > > We sincerely thank you again for these insightful comments. If you have any further suggestions or additional baselines to recommend, we would be happy to consider them.

---

> > > > ### Comment · Reviewer_Pkns · 2024-11-24
> > > >
> > > > Thanks again for the discussion and extensive experiments. I think this will be a good paper after addressing all the reviewers' concerns. I would like to raise the score to 6.

---

> ### Author Response · Authors · 2024-11-22
> **Response to Reviewer Pkns (Part 2/3)**
>
> > W2. "*The toy example offers a controlled setting to assess sequential pattern recognition with the existing methods. However, it may oversimplify the problem, limiting its ability to reflect real-world dynamics. I think the authors should analyze the performance limitations in realistic settings to provide a more comprehensive understanding of the critical limitations of the existing methods, offering insights about the possible improvement areas.*"
>
> We agree that real-world dynamic systems are far more complex, involving intricate dynamics. Therefore, our analysis begins with real-world datasets, as demonstrated in Figure 2, where we separately plot the MRR scores for seen and unseen edge predictions. The results in Figure 2 reveal that while existing methods perform well in predicting seen edges, they struggle significantly with unseen edges. Motivated by this observation, we designed the toy example to illustrate these limitations in a clearer and more straightforward manner.
> We believe that the toy example is valuable for the following reasons:
> 1. This simple example is sufficient to reveal the limitations of existing methods. Due to its straightforward repeating sequential pattern, existing methods are expected to perform well in this scenario. However, all methods fail, providing strong evidence of their inherent limitations.
> 2. The toy example provides a clear and direct illustration of why existing methods fail. Focusing on the simplest case allows us to pinpoint the reasons for their shortcomings. This is detailed in Lines 240-301 of our manuscript, where we analyze the memory module and aggregation module to demonstrate these limitations.
> 3. It provides insights for improvement. The failure of existing methods in the toy example directly stems from their inability to distinguish between $i_4$ and $i_9$. This underscores the need for mechanisms with more robust representation capabilities to distinguish nodes with similar temporal neighborhoods.
>
> Last but not least, we would like to emphasize that our primary contribution lies in the proposed benchmark, which is specifically designed to evaluate the ability of existing temporal GNN methods to predict unseen edges. This addresses a critical limitation of existing benchmarks, which predominantly focus on seen edges. Analyzing existing methods in more carefully designed, realistic settings remains an important future direction, and we plan to explore this in our future work.
>
> > W3. "*The analysis in Tables 3 and 4 provides limited insights, focusing primarily on overall performance decreases with the new benchmark. Deeper insights should be included to inspire future model design for this benchmark. For example, evaluating models separately on seen and unseen edges to highlight how well they handle these distinct cases.*"
>
> Thank you for the suggestion. We present the MRR scores (%) for seen and unseen edges separately in the table below for the Taobao and Yelp datasets. The results demonstrate that most methods perform well on seen edges but struggle with unseen edges. Interestingly, GraphMixer performs slightly better on unseen edges than on seen edges in the Yelp dataset. This may be attributed to GraphMixer's strong generalization capabilities. The low ratio of seen edges in the Yelp dataset likely compels GraphMixer to place greater emphasis on unseen edges, leading to improved performance on unseen edges and relatively lower performance on seen edges. We will include this interesting observation in our revised manuscript.
>
> ||Taobao||Yelp||
> |-|-|-|-|-|
> ||unseen|seen|unseen|seen|
> |GraphMixer|31.97|35.63|31.81|29.58|
> |TCL|35.35|79.03|17.57|49.61|
> |TGAT|30.00|32.85|20.42|22.97|
> |CAWN|35.88|99.12|22.98|90.06|
> |DyGFormer|OOT|OOT|19.50|81.05|
>
> Kindly note that we also provide a performance comparison between our proposed datasets and existing datasets in Table 3. The results demonstrate that the performance of existing methods differs significantly between our datasets and existing benchmarks. For example, DyGFormer and GraphMixer, two of the strongest performers on existing benchmarks, underperform on our datasets compared to other methods. This discrepancy highlights that existing benchmarks may not adequately evaluate the capabilities of temporal GNNs and underscores the necessity of our proposed benchmark.

---

> ### Author Response · Authors · 2024-11-22
> **Response to Reviewer Pkns (Part 3/3)**
>
> > W4. "*The authors claim that they include a state-of-the-art recommendation model named SGNN-HN. However, it was published in 2020, which is not the state-of-the-art recommendation model.*"
>
> Thank you for pointing this out. We agree that SGNN-HN is not the absolute state-of-the-art recommendation model. As stated in our manuscript, SGNN-HN is described as "one of the state-of-the-art methods for sequential recommendation". We will revise this description to "a competitive model designed for sequential recommendation" in our revised manuscript.
> Kindly note that SGNN-HN is included solely for comparison purposes, to demonstrate that it is possible to achieve better performance (as shown by SGNN-HN) on our proposed datasets, whereas existing temporal graph learning models fail to do so. Therefore, SGNN-HN is suitable as a comparative baseline, even though it is not the absolute state-of-the-art.
>
> > Q1. "*The authors only use one metric MRR to measure the performance, what about other ranking metrics (such as NDCG, Recall)? Do we need to design other metrics to measure the performance?*"
>
> Thank you for raising this point. Please note that MRR is widely used in the literature for temporal graph learning [R2], and existing (temporal) graph learning benchmarks also adopt MRR as the **sole** evaluation metric for link prediction [R1, R3]. Therefore, we use MRR as our main evaluation metric.
>
> In this paper, our focus is on proposing benchmark datasets that challenge existing methods with complex sequential dynamics. The MRR results of existing methods on our TGB-Seq datasets effectively demonstrate their limitations in achieving good performance on TGB-Seq. Other metrics are certainly valuable for assessing TGB-Seq datasets and can be chosen by model developers based on the specific requirements of their tasks.
>
> [R1] Temporal graph benchmark for machine learning on temporal graphs. NeurIPS'23.
>
> [R2] Weilin Cong, Si Zhang, Jian Kang, Baichuan Yuan, Hao Wu, Xin Zhou, Hanghang Tong, and Mehrdad Mahdavi. Do we really need complicated model architectures for temporal networks?
>
> [R3] Open graph benchmark: Datasets for machine learning on graphs. NeurIPS'20.

---

### Meta-Review · Area_Chair_7gsa · 2024-12-15

**Metareview:**

The benchmark paper identified an important issue in existing graph datasets (repeated edges). The benchmark is diversified and extensive, spanning multiple domains. The writing and organization is clear and well executed. There is general consensus among the reviewers to accept this paper.

The reviewers also raised some issues, such as additional related work, evaluation setting, the formulation of section 3.2, which have been addressed satisfactorily in the rebuttal phase.

The authors are suggested to go through the issues raised by each reviewer again, and make sure they are addressed in the final version.

**Additional Comments On Reviewer Discussion:**

The reviewers mainly brought additional related work, evaluation settings of this work, and the formulation or organization of section 3.2. The authors have addressed these issues during the rebuttal, and most reviewers have acknowledged that they are satisfied and increased their scores.

---

### Decision · Program_Chairs · 2025-01-22

Accept (Poster)